# Benchmarking universal quantum gates via channel spectrum

Yanwu Gu [1,2] ✉, Wei-Feng Zhuang[1], Xudan Chai[1,2] & Dong E. Liu [1,2,3,4] ✉

Noise remains the major obstacle to scalable quantum computation. Quantum benchmarking provides key information on noise properties and is an important step for developing more advanced quantum processors. However, current benchmarking methods are either limited to a specific subset of quantum gates or cannot directly describe the performance of the individual target gate. To overcome these limitations, we propose channel spectrum benchmarking (CSB), a method to infer the noise properties of the target gate, including process fidelity, stochastic fidelity, and some unitary parameters, from the eigenvalues of its noisy channel. Our CSB method is insensitive to state-preparation and measurement errors, and importantly, can benchmark universal gates and is scalable to many-qubit systems. Unlike standard randomized schemes, CSB can provide direct noise information for both target native gates and circuit fragments, allowing benchmarking and calibration of global entangling gates and frequently used modules in quantum algorithms like Trotterized Hamiltonian evolution operator in quantum simulation.

The performance of today's quantum computers is severely affected by noise and the limited number of qubits[1]. Quantum error correction and fault-tolerant schemes may someday unlock the full potential of quantum computation[2–7], but more precise gate operations must be developed beforehand. It is crucial and necessary to obtain information on the gate noise characteristics and their performance benchmarks in order to calibrate and optimize these gate operations[8–10]. Nonetheless, there is a trade-off between the noise information obtained and the resource overhead for their testing experiments[11]. Process tomography[12,13] is a typical technique for reconstructing the matrix representation of a quantum process, with which the full information of noise is at hand. However, process tomography has exponentially increasing experimental costs and suffers from state-preparation and measurement (SPAM) errors. Although its variant, the gate-set tomography[14–17], can handle SPAM errors, the experimental costs cannot be reduced unless assuming noise models with some properties such as low rank[18].

In reality, for probing noise strength or noise types of a gate, the full reconstruction of the noisy process is not necessary[19,20]. For instance, the average gate fidelity, which measures the average

performance of the implemented noisy gates, can be efficiently obtained by randomized benchmarking (RB)[21–27]. The RB protocol is insensitive to SPAM errors, and its variants[8,28–30] can be applied to benchmark devices with larger system sizes. It is important to note that protocols like RB do not directly measure the fidelity of individual quantum gates but rather the average fidelity of some random circuit fragments[31–33]. To determine the fidelity of a specific target gate (in this paper, we use the phrase "target gate" for any target unitary including a circuit fragment, and later use the phrase "native gate" for a single operational quantum gate), additional strategies, such as an interleaved scheme[34] or altering the sampling distribution of random circuits[28,30], must be incorporated into the modified RB protocol, which can induce more experimental cost and is prone to a large systematic uncertainty[35]. Additionally, to simplify the functional form of measured signals in RB methods, it is often necessary to use group twirling, which limits the types of gates that can be benchmarked. As a consequence, the RB protocols based on random Clifford circuits can only be applied to benchmark the Clifford gates; however, the important non-Clifford gates have to rely on more complicated random circuit sets in which their

[1]Beijing Academy of Quantum Information Sciences, Beijing 100193, China. [2]State Key Laboratory of Low Dimensional Quantum Physics, Department of Physics, Tsinghua University, Beijing 100084, China. [3]Frontier Science Center for Quantum Information, Beijing 100184, China. [4]Hefei National Laboratory, Hefei 230088, China. ✉e-mail: guyw@baqis.ac.cn; dongeliu@mail.tsinghua.edu.cn

native gates belong to other groups instead of Clifford group, e.g., dihedral groups[36,37].

In this work, we introduce channel spectrum benchmarking (CSB), a scalable protocol to estimate the individual noise properties of a universal quantum process from the noisy eigenvalues of its corresponding quantum channel. In CSB protocol, the noisy eigenvalues are first obtained from control-free phase estimation circuits[38–42], which are robust to SPAM errors; and then we establish a connection between the noisy eigenvalues and the diagonal entries of the matrix of pure noise process. From these diagonal entries, we can estimate some noise properties, for example, process fidelity, stochastic fidelity (a quantity similar to unitarity[43,44]), and some important unitary parameters of native gates. We demonstrate the performance of our protocol with certain typical simulated experiments, i.e., 1-qubit Pauli rotation gates, 2-qubit fermionic-simulation (Fsim) gates, 3-qubit circuit fragment implementing Toffoli gate, and 10-qubit circuit fragment implementing an Ising evolution operator. The numerical results show that our CSB protocol can accurately estimate the noise properties.

To give a clearer picture of the performance of our CSB to measure average gate fidelity, in Table 1, we compare our CSB protocol with other leading benchmarking protocols under three aspects: (1) what gates they can benchmark; (2) what type of fidelity they actually measure; and (3) under what conditions they can be scalable to many-qubit systems.

In addition to measuring average gate fidelity, our CSB can also measure the coherence of noise of the target gate. Because the amplitudes of channel eigenvalues are not affected by coherent noise, we can define a quantity called stochastic fidelity to model the strength of stochastic noise only, which is similar to the unitarity[43]. Although unitarity can be measured by purity RB[43] or speckle purity benchmarking[8], both protocols are not scalable. In purity RB, purity measurement has to be performed via measuring all the Pauli operators, which is, however, increasing exponentially with the number of qubits. In speckle purity benchmarking, an exponential number of measurements are required to fully characterize the probability distribution for a given random circuit. The stochastic fidelity in our CSB is, however, scalable because we only need to measure a constant number of noisy eigenvalues of the target gate, which is independent of the system dimension. Moreover, from the phases of noisy eigenvalues, we can measure the actual values of some unitary parameters of the target gate, which gives more specific unitary noise information such that the associated errors can be readily compensated in the experiment. This is a systematic generalization of previous works, for example, robust phase estimation[38] for single-qubit gates and Floquet calibration for Fsim gates[41,45,46].

The CSB protocol can be employed immediately to calibrate quantum gates by using the measured figures of merit as a cost function in the calibration optimization problem[8–10]. Our method can provide more specific information, including process infidelity, stochastic infidelity, and certain key unitary parameters of the target gate under calibration. Additionally, our method can be used to calibrate universal gates, including not only 1 or 2-qubit native gates but also many-qubit native gates such as Mølmer-Sørensen gates[47,48] used in ion trap systems. It may also be interesting to use our method to calibrate certain circuit fragments that are commonly used in quantum algorithms, such as the Trotterized Hamiltonian evolution operator in quantum simulation[45,49–54]. We believe our protocol will pave an important way for the development of cleaner and large-scale quantum devices.

## Results

### Gate fidelity and noisy channel spectrum

We first provide some preliminaries about quantum channels, the fidelity of implemented noisy gates, and the relationship between the fidelity of a gate and the channel spectrum of its noisy implementation.

Consider a quantum gate $U$ acting on a $d$-dimensional space with eigenvalues $e^{i\lambda_a}$ and eigenstates $|\phi_a\rangle$ such that $U|\phi_a\rangle = e^{i\lambda_a}|\phi_a\rangle$. Because of noise, the actual implementation of the gate should be denoted as a quantum channel $\widetilde{\mathcal{U}} = \mathcal{E}\mathcal{U}$, or say completely-positive and trace-preserving (CPTP) map[12], where $\mathcal{U}$ is the corresponding quantum channel of the ideal gate $U$ and $\mathcal{E}$ is a pure noise process. Quantum channels are usually denoted by a set of Kraus operators, for example, $\mathcal{U}(\rho) = U\rho U^\dagger$ and $\mathcal{E}(\rho) = \sum_k E_k \rho E_k^\dagger$ where $\rho$ is an arbitrary operator. Quantum channels can also be represented by a matrix on the basis of $d^2$ dimensional operator space, for example, Pauli operators. We will use the two representations interchangeably and the same symbols for both the abstract quantum channels and their matrix representations.

One can use some fidelity measures to assess the performance of the implemented noisy gate $\widetilde{\mathcal{U}}$, such as process fidelity (also referred to as entanglement fidelity), which is defined as

$$F(\mathcal{U}, \widetilde{\mathcal{U}}) = \mathrm{tr}\left\{\mathcal{I} \otimes \mathcal{U}(|\alpha\rangle\langle\alpha|)\, \mathcal{I} \otimes \widetilde{\mathcal{U}}(|\alpha\rangle\langle\alpha|)\right\} \quad (1)$$

where $|\alpha\rangle = \frac{1}{\sqrt{d}}\sum_{i=1}^d |i\rangle \otimes |i\rangle$ is the maximally entangled state. The process fidelity is closely related to another ubiquitous measure, the average gate fidelity[55]

$$\begin{aligned} F_{\mathrm{ave}}(\mathcal{U}, \widetilde{\mathcal{U}}) &= \int d\psi\, \mathrm{tr}\left\{\mathcal{U}(|\psi\rangle\langle\psi|)\widetilde{\mathcal{U}}(|\psi\rangle\langle\psi|)\right\} \\ &= \frac{dF + 1}{d + 1}. \end{aligned} \quad (2)$$

## Table 1 | Comparison with other leading benchmarking protocols

|  | Gates | Fidelity | Conditions for scalability |
|---|---|---|---|
| CSB | Universal | Target gate | • Eigen-decomposition of target gate is possible<br>• Initial state preparation is efficient |
| Clifford RB[22,23] | Clifford | Ave. among Clifford gates | Not scalable due to compilation issue[28] |
| Mirror RB[30,97] | Universal | Ave. among rand. cycles | Only applicable to gate sets with Clifford gates, arbitrary 1-qubit gates, and 2-qubit controlled Pauli rotations |
| CB[29] | $U^m = I$ | Target + twirling gates | The target gate is Clifford |
| XEB[8] | Universal | Ave. among rand. cycles | Circuits can be classically simulated |

We compare our CSB protocol with other benchmarking protocols under three aspects: (1) what gates they can benchmark; (2) what type of fidelity they actually measure; (3) under what conditions they can be scalable to many-qubit systems. Usually, our CSB measures the fidelity of the target gate except in the case of benchmarking native gates with some type of strong unitary noise, where CSB measures the average fidelity of the compositions of the target gate and twirling gates (for performing randomized compiling[68,69]), see Supplementary Note 3. Our CSB is scalable as long as eigen-decomposition of the target is possible and the number of single and two-qubit gates in the circuits preparing initial states scales at most polynomials with the number of qubits. Clifford RB uses random Clifford circuits to simplify noise and thus only applies to Clifford gates. The fidelity they actually measure is the average of fidelities among random Clifford gates. Mirror RB is initially used to benchmark random cycles generated by Clifford gates and has recently extended to some non-Clifford gates[97]. For cycle benchmarking (CB), the gate or cycle $U$ that can be benchmarked must satisfy $U^m = I$ where $m$ is an integer. CB uses Pauli twirling to simplify noise and thus measures the fidelity of the composition of the target and twirling gates. It needs to compute the output Pauli operators of ideal circuits, which is possible only when the target gate is Clifford for large systems. XEB uses random universal circuits to simplify noise, so it measures the average of fidelities among some random circuit cycles. It requires the classical simulation of circuits to obtain the ideal probabilities of sampled bit strings, which limits its scalability.

It has been proven that the process fidelity only depends on the trace of the pure noise $\mathcal{E}$[55], that is

$$F(\mathcal{U}, \widetilde{\mathcal{U}}) = \frac{\mathrm{tr}\left\{\mathcal{U}^\dagger \widetilde{\mathcal{U}}\right\}}{d^2} = \frac{\mathrm{tr}\{\mathcal{E}\}}{d^2}. \tag{3}$$

Current benchmarking methods, for example, RB and its variants, measure the information of $\mathrm{tr}\{\mathcal{E}\}$ on a basis composed of Pauli operators. In these protocols, Clifford twirling or Pauli twirling are used to simplify the noise matrix $\mathcal{E}$, that is, only diagonal entries of $\mathcal{E}$ on the Pauli basis are kept, such that the relevant figure of merit can be extracted easily from measured signals. The twirling operations need to be performed by running some random circuits. This causes RB-type methods to only apply to benchmark some subsets of quantum gates (e.g., Clifford gates for Clifford RB) and only measure the average fidelity of a set of gates, including both the target gate and the twirling gates.

Instead of focusing on the Pauli operator basis, one can note that the ideal channel $\mathcal{U}$ also induces a natural operator basis composed of its eigen-operators $|\phi_a\rangle\langle\phi_b|$ (corresponding eigenvalues are $e^{i(\lambda_a - \lambda_b)}$). If we can measure the diagonal entries of noise $\mathcal{E}$ in this basis, we can also estimate the gate fidelity. This can be seen from the relationship between the eigenvalues of noisy gate $\widetilde{\mathcal{U}}$ and those of ideal gate $\mathcal{U}$[56], that is

$$g_{ab}e^{i\lambda_{ab}} \approx e^{i(\lambda_a - \lambda_b)} \mathrm{tr}\left\{(|\phi_a\rangle\langle\phi_b|)^\dagger \mathcal{E}(|\phi_a\rangle\langle\phi_b|)\right\} \tag{4}$$

where $g_{ab}$ and $\lambda_{ab}$ is the amplitude and phase of an eigenvalue of $\widetilde{\mathcal{U}}$ with eigen-operator $M_{ab}$, that is $\widetilde{\mathcal{U}}(M_{ab}) = g_{ab}e^{i\lambda_{ab}}M_{ab}$. For the spectrum of quantum channels, there are some useful properties[57]: (1) the eigenvalues lie in the unit disc of complex plain, i.e., $0 \leq g_{ab} \leq 1$ (2) the eigenvalues and eigen-operators always come in conjugate pairs, i.e., for every eigenvalue $g_{ab}e^{i\lambda_{ab}}$ we have $\widetilde{\mathcal{U}}(M_{ab}^\dagger) = g_{ab}e^{-i\lambda_{ab}}M_{ab}^\dagger$.

The relationship Eq. (4) is derived from the first-order perturbation theory under the assumption that noisy gate $\widetilde{\mathcal{U}}$ is diagonalizable[56] (also see Supplementary Note 1). Thus a diagonal entry of $\mathcal{E}$ in the basis composed of $|\phi_a\rangle\langle\phi_b|$ can be obtained

$$\mathcal{E}_{ab,ab} \approx g_{ab}e^{i\lambda_{ab}}e^{-i(\lambda_a - \lambda_b)}. \tag{5}$$

As long as we can measure the noisy eigenvalues $g_{ab}e^{i\lambda_{ab}}$ of $\widetilde{\mathcal{U}}$ and identify their corresponding ideal eigenvalues $e^{i(\lambda_a - \lambda_b)}$, we obtain the diagonal entries of $\mathcal{E}_{ab,ab}$ by Eq. (5). If we can uniformly at random sample some noisy eigenvalues $g_{ab}e^{i\lambda_{ab}}$ or equivalently $\mathcal{E}_{ab,ab}$, then we can use the average of these samples to obtain an estimate of process fidelity $F = \mathrm{tr}\{\mathcal{E}\}/d^2$. Because all the diagonal entries have amplitude smaller than 1, we can prove that the number of samples needed is independent of system dimension from Hoeffding's inequality[58], see "Methods".

Besides the process fidelity, the noisy eigenvalues can also be used to infer the noise strength of stochastic noise only. Since the amplitudes of eigenvalues are only affected by stochastic noise and not changed under unitary noise, we can use those amplitudes to define a quantity referred to as stochastic fidelity

$$F_{\mathrm{sto}} = \sqrt{\frac{1}{d^2}\sum_{ab} g_{ab}^2}. \tag{6}$$

to assess the impact of stochastic noise only.

We can also estimate the actual values of some unitary parameters of a native gate (i.e., unitary errors) from the phases $\lambda_{ab}$ of noisy eigenvalues. This is achieved by identifying the relationship between these unitary parameters and some eigenvalues of the gate, which is similar to the robust phase estimation[38] and Floquet calibration[41,45,46]. We emphasize that, compared to the stochastic errors, the unitary

errors may cause more subtle and complicated problems in quantum error correction and fault-tolerant quantum computation[59–63]. As a result, differentiating between stochastic and unitary errors can assist us in recognizing their respective impacts and in addition, can help to calibrate and tailor the error types.

## The CSB protocol

We now present a practical procedure, which we refer to as CSB, to measure the individual fidelity of a universal process $U$, which can be either a native gate or a circuit fragment.

The estimate of fidelity of the gate $U$ requires a uniform sample of diagonal entries of $\mathcal{E}$, which is identical to a uniform sample of noisy eigenvalues $g_{ab}e^{i\lambda_{ab}}$. The noisy eigenvalues can be estimated by the circuits of control-free phase estimation depicted in Fig. 1. In these circuits, we first prepare state $\rho$, then repeatedly apply the target gate $U$ for $L$ times, and finally measure the expectation value of an operator $O$. We denote the noisy version of $\rho$ and $O$ as $\widetilde{\rho}$ and $\widetilde{O}$. The noisy eigen-operators $M_{ab}$ of $\widetilde{\mathcal{U}}$ can be used as a basis (not necessarily orthonormal) to expand the initial state $\widetilde{\rho}$, that is

$$\widetilde{\rho} = \sum_{ab} \mathrm{tr}\left\{G_{ab}^\dagger \widetilde{\rho}\right\} M_{ab} \tag{7}$$

where $G_{ab}$ is the corresponding left eigen-operator of $M_{ab}$ and they satisfy $\mathrm{tr}\left\{G_{ab}^\dagger M_{a'b'}\right\} = \delta_{ab,a'b'}$. Under the first order perturbation, the noisy eigen-operators $M_{ab}, G_{ab}$ are equal to their corresponding unperturbed eigen-operators $M_{ab}^0, G_{ab}^0$, i.e., the ideal eigen-operators of $\mathcal{U}$, see Supplementary Note 1. For ideal eigen-operators with non-degenerate eigenvalue, we have $M_{ab}^0 = G_{ab}^0 = |\phi_a\rangle\langle\phi_b|$; for ideal eigen-operators with degenerate eigenvalue, the $M_{ab}^0, G_{ab}^0$ are superposition of eigen-operators $|\phi_a\rangle\langle\phi_b|$ in the corresponding degenerate subspace. Then we can show that the expectation value of $O$ at length $L$ under noise is

$$\begin{aligned}\left\langle\widetilde{O}\right\rangle_L &= \mathrm{tr}\left\{\widetilde{O}\widetilde{\mathcal{U}}^L(\widetilde{\rho})\right\} \\ &= \sum_{ab} \mathrm{tr}\left\{\widetilde{O}M_{ab}\right\} \mathrm{tr}\left\{G_{ab}^\dagger \widetilde{\rho}\right\} (g_{ab}e^{i\lambda_{ab}})^L\end{aligned} \tag{8}$$

This is a damping oscillating function. From the time series data $\langle\widetilde{O}\rangle_L$ at different depth $L$, we can extract the noisy eigenvalues via signal processing methods, such as matrix pencil method[64–66]. The imperfect initial state $\widetilde{\rho}$ and measurement operator $\widetilde{O}$ only affect the coefficients of signals rather than the noisy eigenvalues. Thus, the estimate of noisy eigenvalues is insensitive to the SPAM errors as long as SPAM errors are not very large such that the signals incorporating the desired eigenvalues are completely suppressed.

By selecting an appropriate initial state $\rho$ and measurement operator $O$, we can control the number of eigenvalues presented in the resulting signals. The presence of too many different eigenvalues in the signals can pose some difficulties. These include (1) the requirement for a large amount of data or, equivalently, a larger depth $L$ (which is limited by the damping rate $g_{ab}$), and the difficulties in extracting the eigenvalues from the limited measured signals, (2) the difficulties to identify the corresponding ideal eigenvalue for a given noisy counterpart, (3) the difficulties to maintain a uniform sample of the diagonal entries of $\mathcal{E}$. To address these issues, we prepare the initial state and measurement operator as follows:

$$|\psi\rangle = c_a|\phi_a\rangle + c_b|\phi_b\rangle \quad \rho = O = |\psi\rangle\langle\psi| \tag{9}$$

which is a superposition of two eigenvectors only. For this type of initial state and measurement operator, there are only several non-trivial damping oscillating modes, i.e., with a large coefficients $\mathrm{tr}\left\{\widetilde{O}M_{ab}\right\}\mathrm{tr}\left\{G_{ab}^\dagger\widetilde{\rho}\right\}$ in the measured signals $\langle\widetilde{O}\rangle_L$. These non-trivial

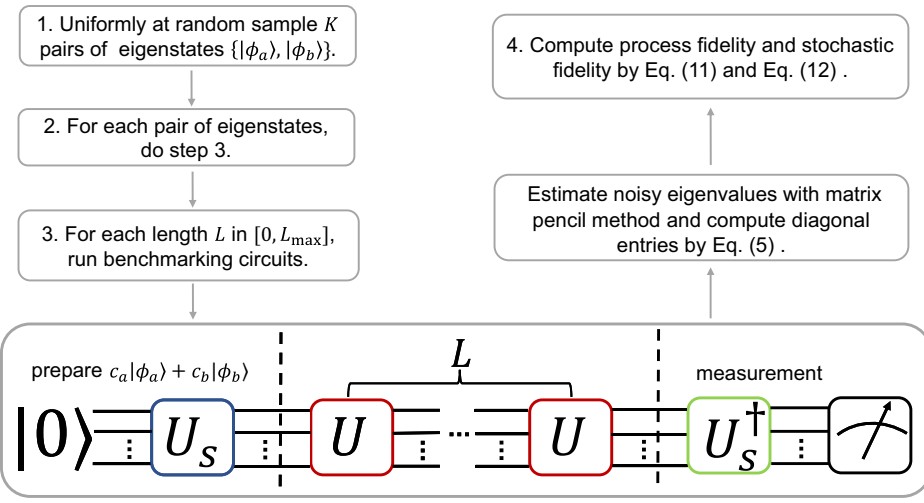

**Fig. 1 | The procedures of channel spectrum benchmarking.** The benchmarking circuits are composed of three parts: the first part $U_s$ prepares the initial state $|\psi\rangle = c_a|\phi_a\rangle + c_b|\phi_b\rangle$, which is a superposition of two eigenstates of target gate $U$; then the target gate $U$ is repeated $L$ times, where $L$ is an integer in $[0, L_{\max}]$; finally, the operator $O = |\psi\rangle\langle\psi|$ is measured. The choice of coefficients $c_a, c_b$ in the initial state is flexible as long as they are comparable and admit an efficient preparation of the initial state. Throughout this work, we choose $c_a = c_b = \frac{1}{\sqrt{2}}$. For each initial state, we estimate several noisy eigenvalues from the time series data $\langle O \rangle_L$ at different depth $L$ using the matrix pencil method.

modes are from the eigen-operators $\{|\phi_a\rangle\langle\phi_b|, |\phi_b\rangle\langle\phi_a|, |\phi_a\rangle\langle\phi_a|, |\phi_b\rangle\langle\phi_b|\}$ shown in the selected initial state and measurement operator.

Thus, as illustrated in Fig. 1, we propose the procedures of CSB below.

1. Uniformly at random sample $K$ pairs of eigenstates $\{|\phi_a\rangle, |\phi_b\rangle\}$ of target unitary operator $U$.
2. For each pair of eigenstates, do step 3, i.e., running phase estimation circuits.
3. In phase estimation circuits, one first prepares the initial state $|\psi\rangle = c_a|\phi_a\rangle + c_b|\phi_b\rangle$, then repeatedly apply the target gate $U$ for $L$ times where $L$ takes successive integers in $[0, L_{\max}]$, finally measure the probability $\langle O \rangle_L$ of obtaining $O = |\psi\rangle\langle\psi|$. Then, we process the measured data using the following steps:
   - 3a. Estimate the noisy eigenvalues $g_{ab}e^{i\lambda_{ab}}$ (amplitudes and phases) from the time series data $\langle O \rangle_L$ by matrix pencil method.
   - 3b. Identify the ideal counterparts of the measured noisy eigenvalues.
   - 3c. Compute the diagonal entries of $\mathcal{E}$ by Eq. (5).
4. Compute the process fidelity by Eq. (11) and stochastic fidelity by Eq. (12).

Step 1 ensures the estimated diagonal entries are uniform samples. We require the amplitude of two coefficients $c_a, c_b$ are comparable and the initial state $|\psi\rangle$ can be efficiently prepared. In the simulated experiments, we always choose $c_a = c_b = \frac{1}{\sqrt{2}}$. The number of initial states $K$ is independent of system dimension $d$ and only depends on desired precision referring to Eq. (18) in "Methods", which is guaranteed by Hoeffding's inequality. So our method is applicable to multi-qubit systems.

In the phase estimation circuits of step 3, we choose the length $L$ from $[0, L_{\max}]$. The maximum length $L_{\max}$ and the number of initial states $K$ determine the total number of benchmarking circuits $N_c = K(L_{\max} + 1)$. In order to collect enough statistics, we need to run each circuit for $N_s$ shots, and therefore, the total experimental cost is

$N_c N_s = K(L_{\max} + 1)N_s$. The choice of $L_{\max}$ and $N_s$ also depends only on the desired precision and not on the system dimension. Previous work has shown that the uncertainty of estimated eigenvalues is inversely proportional to the length $L$[38,41]. Therefore, if higher precision is desired, it is generally better to increase $L_{\max}$ rather than the number of shots $N_s$ per circuit, before the signals are completely degraded.

In step 3a, the noisy eigenvalues are estimated using the matrix pencil (MP) method[64–66]. MP method is well-suited for our task because MP involves a singular value decomposition (svd) of the data Hankel matrix. This svd procedure allows us to keep only the components with non-trivial singular values, i.e., damping oscillating modes caused by noisy eigenvalues of ideal eigen-operators shown in the selected initial state. MP method can reduce some sampling errors and eliminate unwanted eigenvalues (with small coefficients) due to SPAM errors or noisy eigen-operators with degenerate ideal eigenvalue. In our simulated experiments, when using an initial state with unequal phases $\lambda_a, \lambda_b$, the number of obtained noisy eigenvalues is at most four.

In step 3b, our goal is to match the obtained noisy eigenvalues from the MP method to their corresponding ideal counterparts such that we can compute the diagonal entries of $\mathcal{E}$ by Eq. (5). For an initial state, if the two decomposed eigenstates $|\phi_a\rangle, |\phi_b\rangle$ have equal eigenvalues, this process of step 3b is not needed because all ideal channel eigenvalues are 1. On the other hand, if a initial state consists of two eigenstates with unequal eigenvalues, there are three ideal channel eigenvalues $\{e^{i(\lambda_a - \lambda_b)}, e^{-i(\lambda_a - \lambda_b)}, 1\}$ for estimated noisy eigenvalues to match with. To match the obtained noisy eigenvalues to the three ideal ones, we calculate the distance between the phases of the estimated noisy eigenvalues and the ideal eigen-phase $\lambda_a - \lambda_b$ for the corresponding eigen-operator $|\phi_a\rangle\langle\phi_b|$. The noisy eigenvalue with the smallest distance is chosen as the noisy counterpart of the ideal eigenvalue $e^{i(\lambda_a - \lambda_b)}$. Similarly, the noisy counterpart of $e^{-i(\lambda_a - \lambda_b)}$ is also determined. The remaining noisy eigenvalues are considered as the counterparts of the ideal eigenvalue 1. This criterion assumes that the magnitude of the actual phase error $\delta\lambda = \lambda_{ab} - (\lambda_a - \lambda_b)$ is small; more

precisely, we require

$$|\delta\lambda| \ll |\lambda_a - \lambda_b|. \tag{10}$$

If this criterion is not met, which is possibly due to a very large unitary error, we may mismatch the noisy eigenvalues with the ideal ones. Combined with the error mitigation technique for phase estimation in ref. 56, where randomized compiling (RC) is introduced to reduce the phase error (unitary error is transformed to stochastic error, and the total noise strength is not changed), this issue can be fixed.

After calculating the diagonal entries using Eq. (5), we divide them into two categories based on the ideal eigenvalue of the associated basis $|\phi_a\rangle\langle\phi_b|$: one is the trivial operator subspace (dimension $d_{\text{ts}}$) with $\lambda_a = \lambda_b$ (or say the operator subspace spanned by the eigen-operators with eigenvalue 1), the other is the non-trivial operator subspace (dimension $d_{\text{ns}}$) with $\lambda_a \neq \lambda_b$. We should separately compute the average values of diagonal entries in the two subspaces and then combine the two averages to get the estimator of the process fidelity. Because to get a uniform sample of diagonal entries of $\mathcal{E}$, we should assign the sampling probability $\frac{d_{\text{ts}}}{d^2}$ for trivial subspace and probability $\frac{d_{\text{ns}}}{d^2}$ for non-trivial subspace. However, in step 1, we assign the same probability for the two subspaces, that is, the sampling probability $\frac{1}{2}$ for each subspace. The dimension of trivial subspace $d_{\text{ts}}$ is usually very different from the dimension of non-trivial subspace $d_{\text{ns}}$, the probability of sampling an entry in the two subspace are very different. For example, for a many-qubit gate $U$ with non-degenerate operator spectrum, the trivial subspace is spanned by all the eigen-operators with the form $|\phi_a\rangle\langle\phi_a|$, whose dimension $d_{\text{ts}} = d$ is much smaller than $d_{\text{ns}} = d^2 - d$. If there are some degeneracy in the spectrum of the operator $U$, that is $\lambda_a = \lambda_b$ for two different eigenstates $|\phi_a\rangle$, $|\phi_b\rangle$, the trivial subspace can include the eigen-operators of the form $|\phi_a\rangle\langle\phi_b|$. The average value in each subspace can be used to estimate the sum of diagonal entries in the corresponding subspace. Finally, the estimator of the process fidelity is obtained by combining these two averages, that is

$$\hat{F} = \frac{d_{\text{ts}} \overline{\mathcal{E}_{ab,ab}}|_{\lambda_a = \lambda_b} + d_{\text{ns}} \overline{\mathcal{E}_{ab,ab}}|_{\lambda_a \neq \lambda_b}}{d^2} \tag{11}$$

where $\overline{\mathcal{E}_{ab,ab}}$ is the average value of sampled entries. Similarly, the estimator for stochastic fidelity is

$$\hat{F}_{\text{sto}} = \sqrt{\frac{d_{\text{ts}} \overline{g^2_{ab,ab}}|_{\lambda_a = \lambda_b} + d_{\text{ns}} \overline{g^2_{ab,ab}}|_{\lambda_a \neq \lambda_b}}{d^2}}. \tag{12}$$

Our CSB has drawn inspiration from the principles of spectral quantum tomography (SQT)[66]: both methods measure the eigenvalues of the noisy gate. We summarize the differences and the advantages of our CSB compared to SQT as follows.

1. Our CSB is scalable, but SQT is not. First of all, spectral quantum tomography is designed as a method to measure all the eigenvalues of the target gate, which is increasing exponentially with the number of qubits. In our CSB, we only need to measure a limited number of eigenvalues so that we can obtain the most relevant noise information of the target gate, such as process fidelity, stochastic fidelity, and some unitary parameters. This is the primary motivation for all the benchmarking methods instead of doing tomography. Second, the state preparation and final measurement in SQT are on a Pauli basis. Typically, Pauli operators demonstrate a considerable overlap with numerous eigen-operators of the target gate, a factor that results in the signal measured from any given Pauli basis incorporating a multitude of diverse eigenvalues. Therefore, in the context of a system with

high dimensionality, it is infeasible to extract eigenvalues from such a measured signal. Within our CSB methodology, the initial state is selected as a superposition confined to merely two eigenstates of the ideal gate. This choice restricts the number of eigenvalues non-trivially exhibited within the measured signal, thereby facilitating the ease of extracting noisy eigenvalues from the resultant signal.

2. Our CSB gives an accurate estimator for process fidelity using measured noisy eigenvalues, but SQT only gives inequality bounds. We derive a relation between diagonal entries of pure noise channel and noisy eigenvalues of target gate, i.e., Eq. (5), which induces our estimator for process fidelity in Eq. (11). Moreover, we prove that this way to estimate process fidelity can be scalable. The estimate of process fidelity and some unitary parameters also requires the identification of the ideal counter-parts of the measured noisy eigenvalues. This requirement is accomplished via our careful selection of the initial states. Nonetheless, in the context of SQT, all the noisy eigenvalues are concurrently extracted; and therefore, SQT typically presents a challenging task in identifying their corresponding ideal eigenvalues. Consequently, despite the incorporation of our estimator for process fidelity, achieving an accurate estimation with SQT remains a formidable task.

## Numerical simulations with Pauli-rotation gates

We perform simulated experiments to show the performance of our CSB protocol, including single-qubit Pauli rotation gates, two-qubit fermionic-simulation (Fsim) gates, three-qubit Toffoli gate, and an Ising Hamiltonian evolution operator with 10 qubits. Throughout this work, each benchmarking circuit is repeated $N_s = 10^4$ times to collect enough statistics. We will report infidelity (1 − fidelity) instead of fidelity because it's more intuitive to understand the presented results. The error bar of each data point is the standard deviation among the results of ten repetitions of experiments.

Here we measure the infidelity of single-qubit rotation gates, that is

$$R_\sigma(\theta) = e^{-i\frac{\theta}{2}\sigma} \tag{13}$$

where $\theta$ is the rotational angle, and $\sigma$ is a Pauli matrix describing the direction of the rotational axis. This type of unitary operator has two eigenvalues $e^{-i\frac{\theta}{2}}$ and $e^{i\frac{\theta}{2}}$. The dimension of the trivial eigen-operator subspace is 2, which is the same as the dimension of the non-trivial eigen-operator subspace. The corresponding operator (i.e., $\frac{1}{2}(|\phi_a\rangle\langle\phi_a| + |\phi_b\rangle\langle\phi_b|)$) associated with the trivial part of our initial state choice could happen to be very close to one of the noisy eigen-operators of $\widetilde{\mathcal{U}}$. This means that we may only obtain one noisy eigenvalue in this subspace, potentially leading to an inaccurate estimation of the process fidelity. To address this issue, we also prepare another initial state, that is, one of the eigenstates of $R_\sigma(\theta)$ in addition to the superposition state, and then we run phase estimation circuits again for this initial state. Therefore, we have $K = 2$ here. At the same circuit length, we sum the measured probabilities of the two types of circuits (with the two initial states), allowing us to extract all the noisy eigenvalues simultaneously.

Figure 2 shows the results for benchmarking $R_Z(\frac{\pi}{4})$ gate (also known as the $T$ gate). In this simulation, the noise model consists of a combination of stochastic errors (including $T_1$ and $T_2$ errors with equal probabilities $\delta p$) and over/under-rotation errors with angle $\delta\theta$. In Fig. 2a, we fix the unitary error ($\delta\theta = −0.01$) and vary the probability of stochastic error. In Fig. 2b, we fix the stochastic error ($\delta p = 0.001$) and vary the angle of the unitary error. In both cases, we are able to accurately estimate the process and stochastic fidelity of the gate. As a byproduct, we can also estimate the angle of the unitary error by comparing the phases of noisy eigenvalues to their corresponding

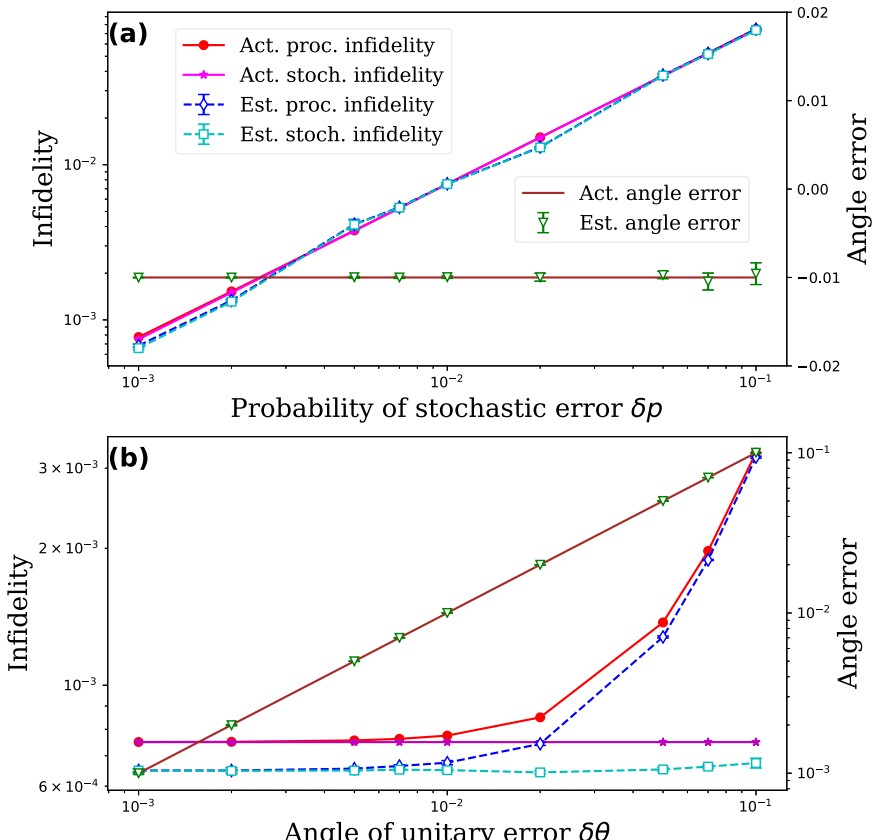

**Fig. 2 | Benchmarking of *T* gate.** In **a**, we fix the unitary error ($\delta\theta = -0.01$) and vary the probability of stochastic error. In **b**, we fix the stochastic error ($\delta p = 0.001$) and vary the angle of the unitary error. The actual process infidelity and stochastic infidelity are obtained by first computing the channel of the noisy gate and then using Eqs. (3) and (6). In both cases; we accurately estimate process infidelity, stochastic infidelity and the angle of unitary error. The accuracy of estimation can be further improved by increasing the circuit length or shots for each circuit.

ideal values. This scheme for unitary error estimation is a more sensitive probe than infidelity measures, as shown in Fig. 2b, where the process infidelity remains almost unchanged when $\delta\theta$ is varied from $10^{-3}$ to $10^{-2}$.

In this simulation, we set $L_{max} = 50$, except when stochastic probability $\delta p = 10^{-3}$, where $L_{max} = 100$. It is worth noting that the accuracy of the estimation can be further improved by increasing the length of the benchmarking circuits. However, increasing $L_{max}$ directly also increases the number of circuits used, which leads to higher costs. Instead, we can repeat the target gate $U$ a certain number of times ($N_{rep}$ times) to create a new target gate, $U' = U^{N_{rep}}$. Correspondingly, the noisy eigenvalue we estimate becomes $(g_{ab}e^{i\lambda_{ab}})^{N_{rep}}$. But remember, we need to determine the ideal eigenvalue from phase difference, thus as a result of Eq. (10), we require

$$N_{rep}|\delta\lambda| \ll |(N_{rep}\lambda_a) \bmod 2\pi - (N_{rep}\lambda_b) \bmod 2\pi|. \qquad (14)$$

**Numerical simulations with Fsim gates**

Here, we benchmark the two-qubit fermionic-simulation (Fsim) gates[8], i.e.,

$$\text{Fsim}(\theta, \phi) = \begin{bmatrix} 1 & 0 & 0 & 0 \\ 0 & \cos\theta & -i\sin\theta & 0 \\ 0 & -i\sin\theta & \cos\theta & 0 \\ 0 & 0 & 0 & e^{i\phi} \end{bmatrix} \qquad (15)$$

where $\theta$ is the iswap angle, and $\phi$ is the control phase angle. We omit some phase parameters that can be freely adjusted by $Z$ rotations.

For the preparation of initial states, we consider all pairs of eigenstates ($K = 6$). The choice of $L_{max}$ is 50 or 100 (for $\delta p = 10^{-3}$). In this simulation, the noise model includes $T_1$, $T_2$ noise with equal probabilities $\delta p$ for all single-qubit gates. For two-qubit gates, each qubit experiences the same errors as single-qubit gates, as well as an over-rotation unitary error with angle errors $\delta\theta$ and $\delta\phi$.

We benchmark a specific Fsim gates with $\theta = \frac{\pi}{4}$, $\phi = \frac{\pi}{2}$, as shown in Fig. 3. In Fig. 3a, we fix the unitary error with $\delta\theta = -0.01$, $\delta\phi = -0.02$, and vary the probability of stochastic error $\delta p$. We accurately estimate all infidelities in this case. However, the estimations of the angles of unitary errors become less accurate when the stochastic error is too strong, as the signal decays too quickly to accumulate enough information to estimate the angles. In Fig. 3b, we fix the probability of stochastic error with $\delta p = 0.001$ and vary the angles of unitary error with $\delta\theta = 0.5\delta\phi = 10^{-3} \sim 10^{-1}$. Again, we accurately estimate all infidelities and angles of the unitary error.

**Numerical simulations with the Toffoli gate**

In this study, we evaluate the performance of the three-qubit Toffoli gate, which is not a native gate but rather a circuit fragment composed of 1-qubit and 2-qubit gates, as shown in Fig. 4c. We randomly select $K = 10$ pairs of eigenstates as the initial state and set $L_{max} = 50$. In the simulated noise model, all single-qubit gates are subject to $T_1$, $T_2$ noise with equal probability $\delta p$. For the two-qubit gates, each qubit experiences the same type of stochastic error as the single-qubit gates, followed by a unitary error of the Fsim type with error angles $\delta\theta = \delta\phi$.

The Toffoli operator has a highly degenerate spectrum, which creates two challenges for our method. First, when sampling noisy eigen-operators, we need them to be uniformly distributed, but for

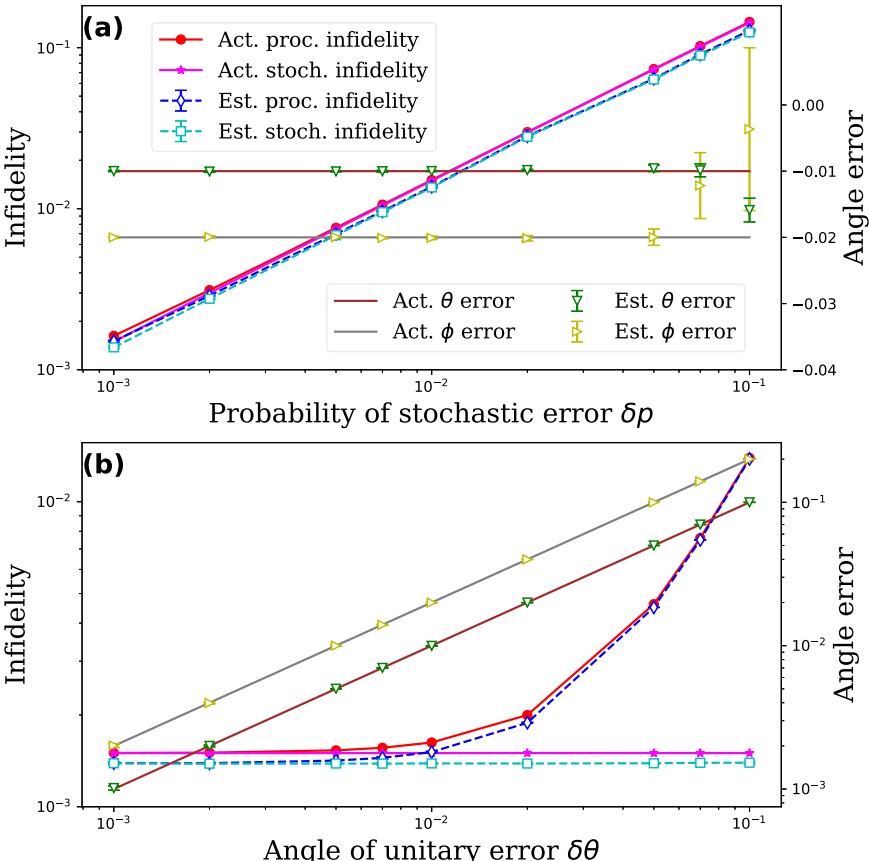

**Fig. 3 | Benchmarking of a Fsim gate with** $\theta = \frac{\pi}{4}$, $\phi = \frac{\pi}{2}$. In **a**, we fix the unitary error with $\delta\theta = -0.01$, $\delta\phi = -0.02$, and vary the probability of stochastic error $\delta p$. In Fig. 3b, we fix the probability of stochastic error with $\delta p = 0.001$ and vary the angles of unitary error with $\delta\theta = 0.5\delta\phi = 10^{-3} - 10^{-1}$. We always accurately estimate the process infidelity and the stochastic infidelity of the gate. However, the accuracy of estimating the angles of the unitary error is compromised when there is a high level of stochastic noise, as the signal degrades quickly and there is not enough data to accurately estimate the angles.

degenerate ideal eigenvalues, the corresponding noisy eigen-operators are superpositions of ideal ones in the degenerate sub-space, which are determined by the details of the noise, see Supplementary Section I. This makes it difficult to generate a uniform sample of noisy eigen-operators. Second, the degenerate eigenvalue may be split by noise into many eigenvalues in the signal, making it harder to extract the noisy eigenvalues, and each eigenvalue may only occupy a small portion of the signal, making them more susceptible to errors. The impact of the highly degenerate spectrum on the estimate of gate noise is demonstrated by the simulated results in Fig. 4a, b.

Usually, some of the degeneracy can be removed by appending a layer of single-qubit gates to the target gate or circuit fragment. For the Toffoli circuit, we append $R_Z(\frac{\pi}{2}) \otimes R_Z(\frac{2\pi}{3}) \otimes R_X(\frac{4\pi}{5})$ to the Toffoli circuit and combine this layer with the last layer of the Toffoli circuit. The choice of the appended layer should keep the state preparation of the new target gate efficient. In the current example, our choice does not change the eigenstates. For the angle parameters in the appended gates, one can design an optimization algorithm to choose the parameters that maximize the distance between eigenvalues. The appended layer of gates results in a varied circuit with a similar structure to the original Toffoli circuit (only the last layer is changed), and they should possess similar noise properties. In the case of strong stochastic error and weak unitary error ($\delta\theta = 0.01$) in Fig. 4a, the benchmarking of the varied circuit provides a very accurate estimate of the process infidelity and the stochastic infidelity of the original Toffoli circuit.

However, there is a significant difference between the estimated and actual process infidelity when the unitary error is very strong, as

shown in Fig. 4b (with fixed stochastic error $\delta p = 0.001$). In Supplementary Note 2, we show that our method may underestimate the process infidelity in the presence of certain strong unitary errors.

One way to address this issue is to introduce random gates into the benchmarking circuits to convert the unitary errors to stochastic errors[67–69]. In Supplementary Note 3, we describe a procedure for transforming noise in the native gates to stochastic errors using random gates from the symmetry group of the target $U$. For benchmarking circuit fragments; we use a technique called RC[68,69] to achieve this. RC is a method that transforms the noise in the circuit into stochastic Pauli errors while maintaining the circuit structure and depth. After RC, the noise type of a circuit cycle is changed, but the process fidelity of the cycle and the circuit structure remain unchanged. As long as there is no repeated structure in $U$ where unitary error can coherently build up and increase the infidelity quadratically with the circuit depth[70] (this is a case where RC should be introduced to suppress the unitary noise), we expect the fidelity of the circuit $U$ to remain unchanged after RC. For each original circuit, we generate $N_r = 10$ random circuits by RC, and each random circuit is run $10^3$ times to keep the cost unchanged. As shown in Fig. 4b, after RC, the varied circuit can accurately estimate the process infidelity of the Toffoli circuit under unitary noise.

### Numerical simulations with Ising evolution operators
Our method is practically scalable if the following two requirements are met:
1. The eigenvalues and eigenvectors of target unitary operator $U$ can be efficiently computed.

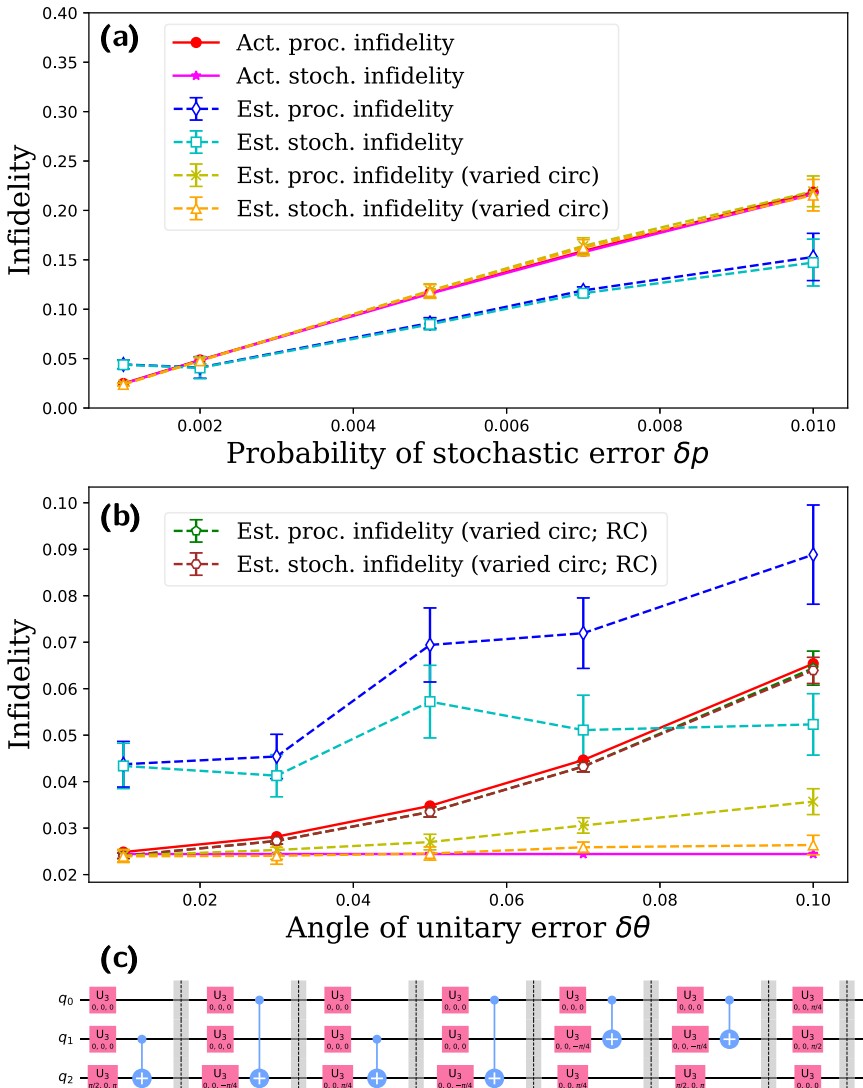

**Fig. 4 | Benchmarking of Toffoli circuit fragment.** We fix the unitary error ($\delta\theta = 0.01$) and vary the stochastic error in (**a**), and fix the stochastic error ($\delta p = 0.001$) and vary the unitary error in (**b**). The circuit implementing the Toffoli gate is presented in (**c**). Due to the highly degenerate spectrum of the Toffoli gate, the estimate of the infidelity is unreliable. However, the degeneracy can be removed by changing the last layer of single-qubit gates. With the varied circuit, we accurately estimate the infidelity of the Toffoli circuit under weak unitary error in (**a**). For strong unitary error, we perform randomized compiling to the benchmarking circuits, converting the unitary error into stochastic error. As a result, the varied circuit also accurately estimates the process infidelity of the Toffoli circuit under strong unitary error, as shown in (**b**).

2. The initial state can be efficiently prepared, i.e., the number of 1-qubit and 2-qubit gates needed for the preparation should, at most, scale polynomials with the number of qubits.

In general, these two requirements are not always satisfied. However, for certain types of unitary operators, such as the evolution operator of an Ising Hamiltonian, these requirements can be met. For an Ising Hamiltonian, the eigenvectors are known and are simply the computational basis states. Given an eigenstate, the eigenvalue can be efficiently computed.

The initial state of a superposition of two computational basis states $|x\rangle = |x_0, \cdots, x_i, \cdots, x_{N-1}\rangle$, $|y\rangle = |y_0, \cdots, y_i, \cdots, y_{N-1}\rangle$ can be prepared as follows: first, for the qubit $i$, if $x_i = y_i$, the state can be prepared by an $X$ gate if $x_i = y_i = 1$; then, for the state of remaining qubits with $x_i \neq y_i$, if we only have one such qubit, a Hadamard gate $H$ can be applied; if there is more than one qubit with $x_i \neq y_i$, one can first prepare a GHZ state on these qubits and then apply some $X$ gates to obtain the target state. Therefore, the preparation of such states costs

at most $N$ 1-qubit and $N$ 2-qubit gates. Additionally, for the evolution operator of the Hamiltonian that can be obtained by performing a local unitary transformation on an Ising Hamiltonian, i.e., $H = \bigotimes_i U_i H_{\text{Ising}} \bigotimes_i U_i^\dagger$, the initial states can also be obtained in a similar way with additional two layers of single-qubit gates $\bigotimes_i U_i, \bigotimes_i U_i^\dagger$. Thus, this type of evolution operator is a good example for benchmarking many-qubit quantum systems.

In the following, we present some important classes of unitary operators frequently used in quantum algorithms or error correction, which are more or less related to the Ising-type of Hamiltonian and satisfy the conditions of scalability of CSB. Thus our CSB is a valuable tool to benchmark these unitary operators and improve their implementation performance by calibration using measured noise information.

- Global entangling gates. Entangling gates are important building blocks for quantum computation. The usual entangling gates are acting only on 2 qubits. Recently, there has been increasing interest in developing global entangling gates based on Ising-

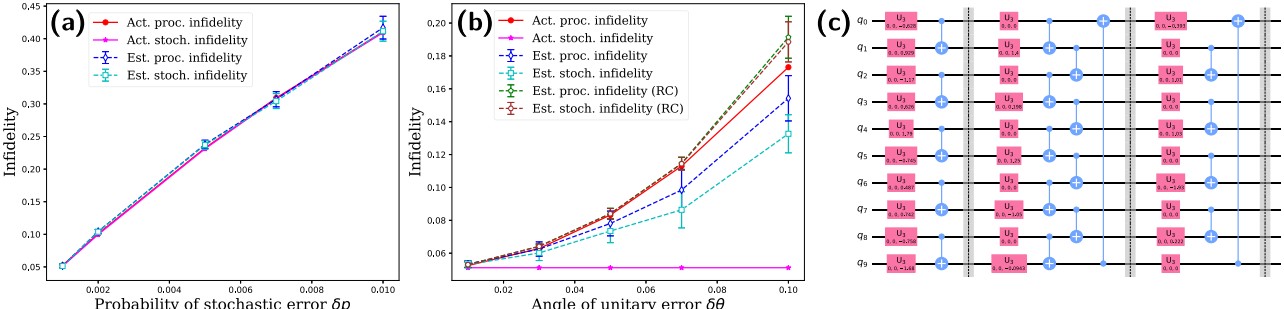

**Fig. 5 | Benchmarking of a 10-qubit Ising evolution operator.** We fix the unitary error ($\delta\theta = 0.01$) and vary the stochastic error in (**a**), and fix the stochastic error ($\delta p = 0.001$) and vary the unitary error in (**b**). The circuit implementing the Ising evolution operator is presented in (**c**). The actual fidelity is not computed from the channel of the circuit but rather inferred from the product of the fidelity of all single-qubit and two-qubit gates. We accurately estimate the process infidelity of the Ising evolution operator under weak unitary error (**a**) and strong unitary error

with RC (**b**). The overestimate of stochastic infidelity in (**b**) is because the unitary error in two-qubit gates is too large for the circuit fragment in (**c**), which causes the presence of many damping oscillating modes in the measured signals. Thus, it is difficult to accurately determine the damping rates. However, the significant differences between the phases of estimated noisy eigenvalues by our method and those of ideal eigenvalues can be used as an indicator of the strong unitary error.

type interactions, which act on multiple qubits or even the whole system. Many works have shown that global entangling gates have a great advantage for circuit compiling compared to the 2-qubit entangling gates[71–74]. These entangling gates have been experimentally realized in Ion trap systems[75–77].

- Cycles in quantum algorithms, such as quantum simulation and quantum optimization. Diagonal unitaries have been applied in simulating chemical dynamics[78], quantum field theories[79,80], and non-unitary evolution[81]. It was also shown that unitary 2-designs, which are useful in device verification and studying complex systems, can be approximately implemented by alternately repeating random unitaries diagonal in the Pauli-Z basis and that in the Pauli-X basis[82]. To simulate a general Hamiltonian $H = \sum_k H_k$, one needs to use the Trotter formula to implement a short time $\Delta t$ evolution of $H$, which is composed of several circuit cycles, each implementing the evolution of a term $e^{-iH_k\Delta t}$. For the efficient implementation of $e^{-iH_k\Delta t}$, each term $H_k$ usually has a locality structure or tensor product structure[12], which causes the unitaries $e^{-iH_k\Delta t}$ satisfy the scalability conditions of CSB. Thus our CSB method can be practically applied to characterize noise in each cycle of the Trotterized Hamiltonian evolution operator. For example, in the Heisenberg model $H = \sum_j J_x \sigma_j^x \sigma_{j+1}^x + J_y \sigma_j^y \sigma_{j+1}^y + J_z \sigma_j^z \sigma_{j+1}^z$, one can characterize the three circuit fragments generated from Pauli-X,Y,Z terms, such as $e^{-i\sum_j J_x \sigma_j^x \sigma_{j+1}^x \Delta t}$, separately by CSB. Similarly, in the QAOA algorithm[83], one can perform CSB separately on the cycles generated by classical Ising interaction and that generated by the transverse field.

- Multiply-controlled gates $C^n(U)$ where $U$ acts only on very few qubits or has a tensor product structure. This class of gates is ubiquitous in quantum error correction[12], Grover's search algorithm[84], and quantum singular transformation[85,86]. One example of this class of gates is the Toffoli gate. One can perform CSB on other $C^n(U)$ within this gate class in a similar manner as we did for the Toffoli gate.

Here we benchmark the evolution operator of a 1-dimensional Ising ring $H = \sum_{i=1}^{10} h_i Z_i + J_{i,i+1} Z_i Z_{i+1}$, where $h_i, J_{i,i+1}$ are randomly chosen. The circuit is shown in Fig. 5c. We sample $K = 10$ pairs of eigenstates and set $L_{max} = 50$. The noise model is the same as that in the benchmarking of Toffoli Gate. The actual process fidelity and stochastic fidelity are inferred from those of single-qubit and two-qubit gates because our computer is not powerful enough to compute the quantum channel of a 10-qubit circuit. Note this procedure of

estimating the fidelity of a circuit from its components is not always reliable[87].

Our method accurately estimates process infidelity under both weak and strong unitary error (with RC), as shown in Fig. 5a, b. The stochastic infidelity in Fig. 5b is over-estimated by our method, which is because the unitary error in the two-qubit gates is too large for the circuit fragment in Fig. 5c. Such large unitary error causes the prepared initial state to have an excessive number of eigen-operators of the noisy target gate, which in turn leads to the presence of too many damping oscillating modes in the measured signals. Consequently, it is difficult to precisely determine damping rates from such complicated signals. However, this strong unitary error can be indicated by the large differences between the phases of estimated noisy eigenvalues and those of ideal eigenvalues in our method.

## Discussion

In this work, we introduced a procedure called CSB, which infers the noise properties of a quantum gate from the eigenvalues of the noisy channels representing the gate. In the protocol, we first choose the initial state using a superposition of a randomly sampled pair of eigenstates of the target gate. Then, we use control-free phase estimation circuits to estimate the noisy eigenvalues in a SPAM error-resistant manner. This choice of initial state simplifies the data processing because the measured signals only contain a few eigenvalues, which can be extracted using signal processing methods such as the MP method. By comparing the noisy eigenvalues to their ideal counterparts, we can estimate noise properties such as the process fidelity, stochastic fidelity, and some unitary parameters of the target gate. Our method can be applied to any quantum gate but performs better on gates with a non-degenerate operator spectrum. For gates with highly degenerate spectrums, we can append a layer of single-qubit gates to remove the degeneracy while maintaining a similar circuit structure. Some types of unitary error can also affect the performance, which can be addressed using randomization techniques like RC. Our method is scalable to many-qubit systems as long as the eigen-decomposition can be computed and the initial state can be efficiently prepared, such as the evolution operator of an Ising-type Hamiltonian.

The requirements for the scalability of our method could be relaxed. In principle, we do not need to obtain the complete set of the eigenmodes for the target gate operator, a few samples of eigenvalues and eigenstates are sufficient. For initial state preparation, there are existing methods for preparing arbitrary states[88–91], but it would be interesting to develop a more efficient algorithm for preparing the particular type of initial states in our method. A variational algorithm[92]

may be able to efficiently prepare these states for most target gates because we have the freedom to choose the coefficients of the superposition states and do not need perfect preparation. Our method can be scaled up in a way similar to simultaneous RB[93,94], where some few-qubit gates are simultaneously benchmarked on different subsets of a many-qubit system such that the effect of crosstalk[95] can be detected.

## Methods

### Number of diagonal entries needed

Here we prove that the number of diagonal entries of pure noise matrix $\mathcal{E}$ needed to estimate process fidelity is independent of system dimension. This proof is based on Hoeffding's inequality: let $X_1, \cdots, X_K$ be independent bounded random variables with $a_i \le X_i \le b_i$ for all $i \in [K]$ and denote their average $\overline{X} = \frac{1}{K}\sum_i X_i$, then for any $\epsilon > 0$ it holds that

$$P\left(\left|\overline{X} - \frac{1}{K}\sum_i \mathbb{E}(X_i)\right| \ge \epsilon\right) \le 2\exp\left(\frac{-2K^2\epsilon^2}{\sum_i(b_i - a_i)^2}\right). \tag{16}$$

This inequality bounds the probability that the empirical average $\overline{X}$ deviates from the average of expectation values of these random variables with a distance $\epsilon$.

Here, we use the average value of some uniformly sampled diagonal entries of pure noise matrix as our estimate of process fidelity. Assume we have $K$ samples of such diagonal entries $\mathcal{E}_{ab,ab}$, so the expectation value of each sampled diagonal entry is $\mathbb{E}(\mathcal{E}_{ab,ab}) = \frac{\text{tr}\{\mathcal{E}\}}{d^2} = F$, and our estimate of the process fidelity is

$$\hat{F} = \frac{1}{K}\sum_{ab} \mathcal{E}_{ab,ab}. \tag{17}$$

Thus, the needed number of diagonal entries $\mathcal{E}_{ab,ab}$ to estimate the process fidelity within an error $\epsilon$ with the probability $1 - \delta$, or say $P(|\hat{F} - F| \le \epsilon) = 1 - \delta$, is

$$K = \frac{\log(2/\delta)}{2\epsilon^2}, \tag{18}$$

which is independent of the system dimension. Here, we take a very conservative bound of $\mathcal{E}_{ab,ab}$, i.e., $0 \le |\mathcal{E}_{ab,ab}| \le 1$. But, the difference between the upper bound and lower bound of $\mathcal{E}_{ab,ab}$ is usually much smaller than 1, so the number of samples needed is much smaller than that in Eq. (18).

## Data availability

The simulated data is available upon request.

## Code availability

The source code for the numerical simulations is available at GitHub repository[96].

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

## Acknowledgements

This work was supported by the Beijing Natural Science Foundation (No. Z220002), the Innovation Program for Quantum Science and Technology (Grant No. 2021ZD0302400), and the National Natural Science Foundation of China (Grant Nos. 12147123 and 11974198).

## Author contributions

Y.G. and D.E.L. wrote the paper. Y.G. and D.E.L. developed the research based on discussions with X.C. and W.Z., and Y.G. and W.Z. performed the simulated experiments. All the authors contribute to discussions of the results and the paper.

## Competing interests

The authors declare no competing interests.
