## [Peer Review File · Nature Communications]

Benchmarking universal quantum gates via channel spectrumReviewers' comments:

Reviewer #1 (Remarks to the Author):

Summary

The authors consider the question of how to test the implementation quality of quantum gates. The main

goal is to overcome known restrictions on the type of gates that can be analyzed with known schemes, e.g.

randomized benchmarking. To this end the authors write down a protocol they call channel spectrum benchmarking (CSB).

The main idea is to compare the known evolution of an eigenstate (or eigenoperator in the channel representation) with

the evolution of such an eigenoperator under the noisy evolution. By applying the unitary channel in question several times

a decay behavior similar to randomized benchmarking can be established. In turn, from this data the corresponding noisy eigenvalue can be extracted via matrix pencil methods in a similar manner as for RB. It is then argued that only a small number of such decay curves have to be sampled in order to estimate the trace of the noise-channel to high precision, which then gives an estimate of the

average fidelity of the implemented channel in comparison to the target unitary channel.

Furthermore, the authors apply their algorithm to data from numerical simulations, based on certain single and two-qubit gates as well as an Ising-Hamiltonian evolution on 10 qubits to showcase its practical functioning.

Assessment

The manuscript considers an important problem within benchmarking of quantum devices and proposes an algorithm that at least

theoretically provides a solution under the given assumptions. However, as the authors note themselves some of these assumptions

are rather restrictive and will in my opinion limit the practical applicability. In particular, it seems to me that at its heart the protocol basically is a variant of interleaved randomized

benchmarking, where the necessity to twirl the error channel with respect to a gate set is defined away by requiring the initial state to be invariant under the original unitary in the first place. However, a lot of protocols such as character-benchmarking or variants thereof are exactly build to avoid the requirement of preparing eigenvectors of a particular operator.

The practicality of the protocol then boils down to the question whether we are mostly interested in testing gates/computations that can be diagonalized on a classical computer beforehand and which will have eigenstates that can actually be prepared efficiently on the device before running the protocol. Although, the authors show the functioning of their protocol for some examples and also for an Ising-Hamiltonian on 10-qubits, where these assumptions are satisfied, I am not convinced that this will be the most relevant scenario. Aside from these more general concerns, the manuscript is well written and the arguments appear to be sound even though I did not check all of them to the last detail. In my opinion however, the assumptions on the error-channel should be spelled out explicitly in the main text. As noted in the appendix, the analysis requires that the channel can be diagonalized - but of course quantum channels could in principle be non-normal.

Based on these considerations, I think the manuscript contains certainly interesting ideas, which would however be better suited for a more specialized journal.

Reviewer #2 (Remarks to the Author):

I enjoyed the paper and found the overall presentation to be quite nice.

As I understand it, the core of the paper is the introduction of channel spectrum benchmarking (CSB). CSB is meant to be a SPAM-robust, scalable benchmarking method for arbitrary quantum gates (or channels). CSB works by estimating quantities of interest, such as process fidelity, by learning a limited number of the eigenvalues of the noisy implementation of the chosen quantum gate. The authors' use Hoeffding's inequality to prove that the number of necessary eigenvalues is independent of system size.

The noisy channel's eigenvalues are learned by running control-free phase estimation circuits, which already exist in the literature. Control-free phase estimation circuits work by preparing an equal superposition state of two eigenvectors of the target quantum gate. The noisy gate is then repeated many times, and the final quantum state is measured against the initial state.

The requirement to prepare an initial superposition of the target gate means that this technique is only scalable when: (i) there is an efficient eigen-decomposition of the target gate and (ii) the initial state preparation is efficient. My understanding is that these conditions are not typically met, which limits the potential impact of the technique.

After outlining how CSB works, the authors go over several case studies in which CSB is used to benchmark gates performed on increasingly large systems.

What follows is a series of: (i) major points; (2) minor points; and (iii) errata that the authors may wish to address.

Major Point 1: CSB bears a striking resemblance to spectral tomography. Both methods use signal processing techniques to learn the eigenvalues of the noisy quantum gate. The main approach is to learn the diagonal elements of the noisy quantum gate in some useful basis. CSB uses the target gate's eigenbasis, while spectral tomography uses the Pauli basis. The circuits used are then equivalent in form (state prep, repeat the gate, measure appropriately). Given the close overlap, I suggest including an in-depth comparison between the two techniques in the paper.

The key questions to answer are: What can this technique do that spectral tomography cannot? Why can you not modify the spectral tomography procedure (say by running a limited selection of carefully chosen circuits) to achieve the same results as this paper? Simply citing the original spectral tomography paper as a reference for matrix pencil methods is not sufficient.

Major Point 2: The paper claims to restrictions on the scalability of CSB. It is not obvious how often these two restrictions apply. Additional discussion of when efficient eigen-

decompositions exists and when state prep is efficient would help clarify the impact of this work.

Minor Point 1: I would appreciate additional arguments for why

$U^L(M_{ab}) = g_{ab}e^{i\lambda_{ab}L}$. It is clear from the first-order perturbation theory argument that this is approximately true when $L=1$. What happens if applying U repeatedly amplifies the contribution from the higher-order terms?

Minor Point 2: Mirror RB exists for universal gatesets (<https://arxiv.org/abs/2207.07272>).

Minor Point 3: The authors' state that

"Although its variant, the gate-set tomography, can handle SPAM errors, the experimental costs cannot be reduced."

This is certainly true in the presence of arbitrary noise. However, it is not obviously true if you assume some sort of reduced noise model.

Minor Point 4: In the supplementary material, the author's refer to the "symmetric group of U ." I am unsure of what this refers to. The provided example looks to be the centralizer of U (although I am often wrong!).

Errata 1: On page 2, replace "MS" with "Mølmer-Sørensen."

Errata 2: On page 2, there is some ambiguity in what the authors' mean when they state "rho is an arbitrary operator." Perhaps "rho is an arbitrary density matrix" is clearer?

Errata 3: On page 3, I am not sure what is meant by "Distracting from the Pauli operator basis."

RESPONSE TO REFEREE A

We divide the report of Referee A into six items (A1 to A6) and respond to them item by item.

A1. The authors consider the question of how to test the implementation quality of quantum gates. The main goal is to overcome known restrictions on the type of gates that can be analyzed with known schemes, e.g. randomized benchmarking. To this end the authors write down a protocol they call channel spectrum benchmarking (CSB). The main idea is to compare the known evolution of an eigenstate (or eigenoperator in the channel representation) with the evolution of such an eigenoperator under the noisy evolution. By applying the unitary channel in question several times a decay behavior similar to randomized benchmarking can be established. In turn, from this data the corresponding noisy eigenvalue can be extracted via matrix pencil methods in a similar manner as for RB. It is then argued that only a small number of such decay curves have to be sampled in order to estimate the trace of the noise-channel to high precision, which then gives an estimate of the average fidelity of the implemented channel in comparison to the target unitary channel. Furthermore, the authors apply their algorithm to data from numerical simulations, based on certain single and two-qubit gates as well as an Ising-Hamiltonian evolution on 10 qubits to showcase its practical functioning.

Response: We thank Referee A for summarizing our work. We are also grateful for all the questions from Referee A. They greatly help the improvement of our manuscript.

A2. The manuscript considers an important problem within benchmarking of quantum devices and proposes an algorithm that at least theoretically provides a solution under the given assumptions. However, as the authors note themselves some of these assumptions are rather restrictive and will in my opinion limit the practical applicability. In particular, it seems to me that at its heart the protocol basically is a variant of interleaved randomized benchmarking, where the necessity to twirl the error channel with respect to a gate set is defined away by requiring the initial state to be invariant under the original unitary in the first place. However, a lot of protocols such as character-benchmarking or variants thereof are exactly build to avoid the requirement of preparing eigenvectors of a particular operator.

Response: We thank Referee A for considering our research content as an important problem.

The assumptions for scalability that we made are: (1) Eigen-decomposition of target gate is possible; (2) Initial state preparation is efficient. For target gates with a few qubits, the eigen-decomposition is an easy task and the circuit for preparing the initial state can be generated by existing software, for example, Qiskit. Thus we believe there will be no problem to apply CSB on few-qubit gates. For many-qubit gates, every protocol has to rely on some assumptions, because quantum benchmarking in its essence is a comparison between noisy results and ideal results which are of course not always possible by classical computation. We have shown our CSB is at least practically scalable for unitary operators generated by the Ising-type of Hamiltonian. This class of unitary operators has very important applications in building global entangling gates, in quantum algorithms, such as quantum simulation and QAOA, and in quantum error correction. We discuss these applications in detail in response to A3. Thus our CSB method can be used in this task to benchmark relevant gates or circuit cycles to improve their performance.

Except for the same goal that we want to estimate the average fidelity of a particular gate or cycle, our CSB method is essentially different from interleaved RB. In the following, we will discuss the requirements or limitations of RB-type of protocols including interleaved RB and character RB, and compare them with our CSB method to show the advantages of CSB.

- RB requires a benchmarking group with very good properties. In RB methods, one first needs to determine a benchmarking group such that the noise channels can be greatly simplified under group twirling, and finally the measured signal is a linear combination of exponential decay functions. These benchmarking groups must possess some very good properties as follows for the practical applicability of RB-type of methods.

1. There must exist an efficient method to uniformly sample gate elements from the benchmarking group in order to perform group twirling. Thus, the benchmarking group is usually a discrete group. But even for discrete groups, uniform sampling may not be efficient if the number of group elements increases exponentially with the number of qubits. Additionally, for any element g of a finite group, there always exists a positive integer r such that $g^r = I$, where I is the identity operator. This means the phase λ_n of any eigenvalue of g is a rational multiple of 2π , i.e., $\lambda_n = \frac{n}{r}2\pi$ where n are integers. The gates or unitary operators which have eigenvalues of such form are of course a very small portion among all unitary operators.
2. The number of non-trivial irreducible representations of the benchmarking group must be small. The number of non-trivial irreducible representations determines how many decay parameters are in the measured signal. Too many decay parameters will cause difficulty to extract their values from the measured signal. Character randomized benchmarking (we cite it as Ref. [27] in the current manuscript) is a method to isolate a particular decay parameter by adding some random gates from a character group (a subgroup of the benchmarking group) to the circuits. The random gates from the character group combined with their characters under an irreducible representation form a projector onto this representation such that one decay parameter associated with this representation is singled out. However, the scalability of character RB induces some limits on the underlying benchmarking group and character group: the number of exponential decays caused by the benchmarking group is required not grow too rapidly with the number of qubits; the dimension of the representation of the character group being projected on is required not grow too rapidly with the number of qubits.
3. The inverse of a sequence of random gates must be efficiently computed. Because the circuits of RB-type of methods should be equal to identity ideally such that we can easily compare noisy results with ideal results. Of course, not all groups have such efficient simulation algorithms.
4. Any group element should be efficiently decomposed into single-qubit gates and two-qubit gates. For example, an element from a n -qubit Clifford group can be decomposed into a sequence of $O(n^2)$ one- and two-qubit gates. However, as the number of qubits increases, the infidelity of these compiled group elements grows rapidly, rendering Clifford RB protocols impractical for relatively small n , even with state-of-the-art gates. To the best of our knowledge, experiments about Clifford RB only involve up to 3 qubits. One way to avoid this limitation is to use a non-uniform sample distribution to form random circuits, such as direct RB (Ref. [28]), Mirror RB (Ref. [30]), and XEB (Ref. [8]). These methods serve as good benchmarks for the holistic performance of quantum devices, but they can not be used to obtain the average fidelity of a particular group element or cycle. Cycle benchmarking (Ref. [29]) allows obtaining the average fidelity for a particular cycle. In cycle benchmarking (CB), one needs to compute the output Pauli operators of ideal circuits. So the scalability for CB seems to be possible only for Clifford cycles. However, for other cycles beyond Clifford structure, it is still unclear whether it's scalable.

The groups with the above properties are quite limited. So, RB is an excellent method to benchmark Clifford gates and some particular non-Clifford gates. But, for a general unitary operator, finding a benchmarking group with the above properties is definitely not an easy task. However, our CSB method does not rely on the good properties of a benchmarking group and provides a general framework to benchmark more general gates in principle.

- RB cannot directly measure the average fidelity of individual gates. To determine the fidelity of a specific target gate, interleaved RB needs to be performed. For interleaved RB, one first needs to

perform a regular RB experiment to obtain the average fidelity of the gates in the benchmarking group as a reference, and then one needs to run another RB experiment where each random gate is combined with the target gate. The measured average fidelity in the experiment with the target gate divided by the reference fidelity gives an estimate of the fidelity of the target gate. The limitations of the regular RB above of course also apply to interleaved RB. The more serious problem of interleaved RB is that the estimate of target gate fidelity is based on an approximation, that is, the group twirling of the composition of the noise channel of the target gate \mathcal{E}_t and the noise channel of reference gates \mathcal{E} is equal to the product of twirling of each individual noise channel. This problem causes interleaved RB only gives bounds on the target gate fidelity as shown in Ref. [35]

$$\left| p(\mathcal{E}_t) - \frac{p(\mathcal{E}_t\mathcal{E})p(\mathcal{E})}{u(\mathcal{E})} \right| \leq \sqrt{1 - \frac{p(\mathcal{E})^2}{u(\mathcal{E})}} \sqrt{1 - \frac{p(\mathcal{E}_t\mathcal{E})^2}{u(\mathcal{E})}} \quad (1)$$

where $p(\mathcal{E}_t\mathcal{E}), p(\mathcal{E})$ are decay parameters obtained in the interleaved RB and regular RB, respectively (decay parameter is linearly related to average gate fidelity). $u(\mathcal{E})$ is the unitarity of reference gates measured by purity RB. From Eq. (1), interleaved RB gives an accurate estimate of target gate fidelity only when \mathcal{E} is depolarizing noise, i.e., $p(\mathcal{E})^2 = u(\mathcal{E})$. Otherwise, the estimate of target fidelity is prone to large systematic uncertainty.

Our CSB method does not need the reference group twirling to simplify noise and thus can directly measure the average fidelity of the target gate. In some particular strong unitary noise, our CSB method may underestimate the average error rate and we need randomized compiling to improve the estimation accuracy. When the twirling gates in randomized compiling cannot be absorbed into the target gates, our method will have a similar issue as interleaved RB. But for our CSB there is greater flexibility to choose the twirling group as we do not need the good properties of group twirling as in the interleaved RB and the target gate does not need to be in the twirling group (see the supplementary Sec. III). So we can choose a group of gates whose error is much smaller than the target gate such that the extra error introduced by twirling gates can be neglected.

- RB cannot directly measure the information of coherent noise. Because of the group twirling used in RB experiments, all the different kinds of noise are transformed into stochastic noise. One needs to perform variants of RB, such as purity RB (Ref. [46]) or XEB (speckle purity benchmarking, Ref. [8]) to measure the coherence of noise in terms of unitarity. However, both protocols are not scalable. In purity RB, purity measurement has to be performed via measuring all the Pauli operators which is however increasing exponentially with the number of qubits. In speckle purity benchmarking, an exponential number of measurements are required to fully characterize the probability distribution for a given random circuit. Furthermore, both methods measure the unitarity of a gate set instead of an individual gate.

The measured signal in our CSB protocol is not a pure exponential decay function as that in RB type of experiments but a decay oscillating function. The oscillating features of the signal are one of the essential differences between our CSB and other protocols of RB type. From the oscillating frequencies of the signal (the phases of the noisy eigenvalues of the target gate) and the decay rates of the signal (the amplitudes of noisy eigenvalues), we can distinguish the unitary noise and stochastic noise in the target gate. Because the amplitudes of channel eigenvalues are only affected by stochastic noise, we can define a quantity called stochastic fidelity to model the strength of stochastic noise, which is similar to the unitarity. The stochastic fidelity in our CSB is however scalable because we only need to measure a constant number of noisy eigenvalues of the target gate, which is independent of the system dimension. Moreover, from the phases of noisy eigenvalues, we can measure the actual values of some unitary parameters of the target gate, which give more specific unitary noise information such that the associated errors can be readily compensated in the experiment. This work represents a comprehensive and enhanced advancement over previous studies, for example, robust phase estimation (Ref. [41]) for single-qubit gates and Floquet calibration for Fsim gates (Refs. [44,48,49]).

We have added a paragraph in the introduction at page 2 to highlight this important point, which was briefly presented in the caption of Table. I of the previous manuscript.

In summary, our CSB method has advantages over existing benchmarking methods in terms of general applicability to many quantum gates, the ease and accuracy to measure the fidelity of individual gate, as well as the measurement of information about unitary noise.

A3. The practicality of the protocol then boils down to the question whether we are mostly interested in testing gates/computations that can be diagonalized on a classical computer beforehand and which will have eigenstates that can actually be prepared efficiently on the device before running the protocol. Although, the authors show the functioning of their protocol for some examples and also for an Ising-Hamiltonian on 10-qubits, where these assumptions are satisfied, I am not convinced that this will be the most relevant scenario.

Response: We thank Referee A for pointing out this important question. We apologize that we did not discuss the applications of the unitary operator generated by the Ising-type of Hamiltonian where our CSB is practically scalable. In the following, we present some important classes of unitary operators frequently used in quantum algorithms or error correction, which are more or less related to the Ising-type of Hamiltonian and satisfy the conditions of scalability of CSB. Thus our CSB is a valuable tool to benchmark these unitary operators and improve their implementation performance by calibration using measured noise information.

- Global entangling gates. Entangling gates are important building blocks for quantum computation. The usual entangling gates are acting only on 2 qubits. Recently, there are increasing interest in developing global entangling gates based on Ising-type interactions, which act on multiple qubits or even the whole system. Many works have shown that global entangling gates have a great advantage for circuit compiling compared to the 2-qubit entangling gates [1–4]. These entangling gates have been experimentally realized in Ion trap systems [5–7].
- Cycles in quantum algorithms, such as quantum simulation and quantum optimization. Diagonal unitaries have been applied in simulating chemical dynamics [8], quantum field theories [9, 10], non-unitary evolution [11]. It was also shown that unitary 2-designs, which are useful in device verification and studying complex systems, can be approximately implemented by alternately repeating random unitaries diagonal in the Pauli-Z basis and that in the Pauli-X basis [12]. To simulate a general Hamiltonian $H = \sum_k H_k$, one can use the Trotter formula to implement a short time Δt evolution of H , which is composed of several circuit cycles each implementing the evolution of a term $e^{-iH_k\Delta t}$. For the efficient implementation of $e^{-iH_k\Delta t}$, each term H_k usually has locality structure or tensor product structure [13], which causes the unitaries $e^{-iH_k\Delta t}$ satisfy the scalability conditions of CSB. Thus our CSB method can be practically applied to characterize noise in each cycle of the Trotterized Hamiltonian evolution operator. For example, in the Heisenberg model $H = \sum_j J_x \sigma_j^x \sigma_{j+1}^x + J_y \sigma_j^y \sigma_{j+1}^y + J_z \sigma_j^z \sigma_{j+1}^z$, one can characterize the three circuit fragments generated from Pauli-X, Y, Z terms, such as $e^{-i \sum_j J_x \sigma_j^x \sigma_{j+1}^x \Delta t}$, separately by CSB. Similarly, in the QAOA algorithm [14], one can perform CSB separately on the cycle generated by classical Ising interaction and that generated by the transverse field.
- multiply-controlled gates $C^n(U)$ where U acts only on very few qubits or has tensor product structure and n is the number of control qubits. For the cases with this special structure, the unitary U is easy to be diagonalized as $U = V\Lambda V^\dagger$, where V acts on the same qubits as U and Λ is a diagonal matrix. Thus the control gate $C^n(U)$ is similar to a diagonal gate by a local unitary $I \otimes V$, which satisfies the scalability conditions of CSB. This class of gates is ubiquitous in quantum error correction [13], Grover’s search algorithm [15], and quantum singular transformation [16, 17]. One example of this class of gates is the Toffoli gate. One can perform CSB on this class of gates in a similar manner as we did for the Toffoli gate.

We have added this discussion in the section “Ising Hamiltonian evolution” at page 10.

A4. Aside from these more general concerns, the manuscript is well written and the arguments appear to be sound even though I did not check all of them to the last detail.

Response: We thank Referee A for considering our manuscript well written and our arguments sound.

A5. In my opinion however, the assumptions on the error-channel should be spelled out explicitly in the main text. As noted in the appendix, the analysis requires that the channel can be diagonalized - but of course quantum channels could in principle be non-normal.

Response: We thank the referee’s suggestion. We have explicitly expressed the assumption on the noisy channel above Eq. (5) on page 3 in the main text.

Our assumption on the noisy channel is only that it can be diagonalizable. But the requirement of being diagonalizable is a much weaker assumption than being normal. Normal channel is a special class of diagonalizable channels, whose eigen-operators form an orthonormal basis for the operator space. The property of orthonormal basis is however not needed in our derivation. There are of course non-diagonalizable channels in special cases. But diagonalizable matrices are dense in the space of all matrices, meaning that any non-diagonalizable matrix can be deformed into a diagonalizable one by a small perturbation. Put differently, the subset of non-diagonalizable matrix has measure zero in the matrix space. Thus, in practice, there seems to be no probability that we can encounter a non-diagonalizable channel.

A6. Based on these considerations, I think the manuscript contains certainly interesting ideas, which would however be better suited for a more specialized journal.

Response: We thank Referee A for considering our ideas interesting.

In response to A2 and A3, we have shown that our CSB has significant advantages over existing benchmarking methods and is practically scalable for some important classes of unitary operators. Thus we believe our work is generally interesting in many directions of quantum computation.

RESPONSE TO REFEREE B

We divide the report of Referee B into twelve items (B1 to B12), and respond to them item by item.

B1. I enjoyed the paper and found the overall presentation to be quite nice.

Response: We are glad to see Referee B enjoyed our paper and considers our presentation to be quite nice.

B2. *As I understand it, the core of the paper is the introduction of channel spectrum benchmarking (CSB). CSB is meant to be a SPAM-robust, scalable benchmarking method for arbitrary quantum gates (or channels). CSB works by estimating quantities of interest, such as process fidelity, by learning a limited number of the eigenvalues of the noisy implementation of the chosen quantum gate. The authors' use Hoeffding's inequality to prove that the number of necessary eigenvalues is independent of system size.*

The noisy channel's eigenvalues are learned by running control-free phase estimation circuits, which already exist in the literature. Control-free phase estimations circuits work by preparing an equal superposition state of two eigenvectors of the target quantum gate. The noisy gate is then repeated many times, and the final quantum state is measured against the initial state.

Response: We thank Referee B for summarizing our work. We are also grateful for all questions from Referee B. These questions help a lot in the improvement of our manuscript.

B3. *The requirement to prepare an initial superposition of the target gate means that this technique is only scalable when: (i) there is an efficient eigen-decomposition of the target gate and (ii) the initial state preparation is efficient. My understanding is that these conditions are not typically met, which limits the potential impact of the technique.*

After outlining how CSB works, the authors go over several case studies in which CSB is used to benchmark gates performed on increasingly large systems.

What follows is a series of: (i) major points; (2) minor points; and (iii) errata that the authors may wish to address.

Response: Indeed, the conditions for scalability are not generally met. As shown in Table I, every benchmarking method is scalable under some restrictions. This seems to be unavoidable for all the schemes because quantum benchmarking is essentially a comparison between noisy results from experiments and ideal results from classical computation which is not always possible for highly dimensional systems. We have shown that our CSB is practically scalable for the class of unitary operators generated by Ising type of interactions, which has many important applications in many directions of quantum computation as shown in the response to A3 (reply to Referee A).

B4. *Major Point 1: CSB bears a striking resemblance to spectral tomography. Both methods use signal processing techniques to learn the eigenvalues of the noisy quantum gate. The main approach is to learn the diagonal elements of the noisy quantum gate in some useful basis. CSB uses the target gate's eigenbasis, while spectral tomography uses the Pauli basis. The circuits used are then equivalent in form (state prep, repeat the gate, measure appropriately). Given the close overlap, I suggest including an in-depth comparison between the two techniques in the paper. The key questions to answer are: What can this technique do that spectral tomography cannot? Why can you not modify the spectral tomography procedure (say by running a limited selection of carefully chosen circuits) to achieve the same results as this paper? Simply citing the*

original spectral tomography paper as a reference for matrix pencil methods is not sufficient.

Response: We thank Referee B for this important question. We have added an in-depth comparison between our CSB and spectral quantum tomography (SQT) (Ref. [69]) below Eq. (12) at page 6: “**Our CSB has drawn inspiration from the principles of the spectral quantum tomography (SQT) [69]**”. The differences and advantages of our CSB compared to spectral quantum tomography are as follows.

1. Our CSB is scalable but spectral quantum tomography is not. First of all, spectral quantum tomography seems to be designed as a method to measure all the eigenvalues of the target gate, which is increasing exponentially with the number of qubits. In our CSB, we only need to measure a limited number of eigenvalues such that we can obtain the most relevant noise information of the target gate, such as the process fidelity, stochastic fidelity, and some unitary parameters. This is the primary motivation for all the benchmarking methods instead of doing tomography.

The inherent reason for the non-scalability of SQT is due to its choice of initial states for the circuits, that is, an eigen-state of Pauli operators. In general, the Pauli operators have non-negligible overlap with many eigen-operators of the target gates, which causes that the measured signal from any Pauli basis will contain many different eigenvalues. Thus, for a high dimensional system, it’s impossible to extract the eigenvalues from such measured signal and one cannot transform SQT to a scalable benchmarking protocol by running a limited selection of circuits. In our CSB, we consider the eigenstates of the ideal target gate, and choose the initial state as a superposition of only two of those eigen-states. Our method limits the number of eigenvalues non-trivially presented in the measured signal, such that we can easily extract noisy eigenvalues from signal.

2. Our CSB gives an accurate estimator for process fidelity using measured noisy eigenvalues but SQT only gives inequality bounds.

We derive a relation between diagonal entries of pure noise channel and noisy eigenvalues of target gate, i.e. Eq. (5), which induces our estimator for process fidelity in Eq. (11). Moreover, we prove that this way to estimate process fidelity can be scalable. The estimate of process fidelity and some unitary parameters also requires the identification of the ideal counterparts of the measured noisy eigenvalues. This requirement is accomplished via our careful selection of the initial states. Nonetheless, in the context of SQT, all the noisy eigenvalues are concurrently extracted; and therefore, SQT typically presents a challenging task in identifying their corresponding ideal eigenvalues. Consequently, despite the incorporation of our estimator for process fidelity, achieving an accurate estimation with SQT remains a formidable task.

Finally, we think our CSB serves as a comparative metric in line with established schemes such as Randomized Benchmarking (RB), RB-like protocols, Cycle Benchmarking, and Cross-Entropy Benchmarking (XEB), among others. For a more comprehensive comparison, if necessary, we direct Referee B to our response in Reply A2 (addressing Referee A’s comments).

B5. Major Point 2: The paper claims to restrictions on the scalability of CSB. It is not obvious how often these two restrictions apply. Additional discussion of when efficient eigen-decompositions exist and when state prep is efficient would help clarify the impact of this work.

Response: We thank Referee B for this suggestion. In response to A3 (addressing Referee A’s comments), we have given some classes of unitary operators which satisfy the scalability conditions of CSB and have very important applications. The shared feature of these unitary operators is that they are similar to a unitary operator generated by a classical Ising Hamiltonian by local unitary transformations. A brief discussion is as follows.

For a target unitary operator U , we assume it has already been diagonalized by a unitary operator V , that is $U = V\Lambda V^\dagger$, where Λ is a diagonal matrix containing the eigenvalues of U and the columns of V are the

eigenstates of U . The diagonal matrix Λ can be considered as a unitary operator generated by a classical Ising Hamiltonian, on which we already know how to perform CSB. This classical Ising Hamiltonian should only contain a polynomial number of terms for efficient computation of eigenvalues given a computational basis state $|i\rangle$ (these terms of the Hamiltonian could be non-local). Then given an eigenstate of U , that is $|\phi_i\rangle = V|i\rangle$, one can efficiently compute its eigenvalue. For the initial state preparation, that is a superposition of two eigenstates $\frac{1}{\sqrt{2}}(|\phi_i\rangle + |\phi_j\rangle)$, one can first prepare $\frac{1}{\sqrt{2}}(|i\rangle + |j\rangle)$ (we have shown how to prepare it in the main text), then apply the circuit implementing V . Thus, for efficient preparation of the initial states in CSB, we require V to have an efficient circuit implementation. An obvious example of V is a tensor product of local unitary operators. For the V with this simple structure, it usually should be easy for us doing the eigen-decomposition in the first place. There may exist some other classes of V that induce a unitary operator U satisfying the scalability conditions of CSB, but this will be future work.

B6. Minor Point 1: I would appreciate additional arguments for why $U^L(M_{ab}) = (g_{ab}e^{i\lambda_{ab}})^L M_{ab}$. It is clear from the first-order perturbation theory argument that this is approximately true when $L=1$. What happens if applying U repeatedly amplifies the contribution from the higher-order terms?

Response: Here we do not use any approximation. By definition, M_{ab} is an eigen-operator of the noisy gate \tilde{U} (we guess the use of U in B6 is a typo of Referee B?) with eigen-value $g_{ab}e^{i\lambda_{ab}}$. Thus we have $\tilde{U}(M_{ab}) = g_{ab}e^{i\lambda_{ab}} M_{ab}$. Then it's easy to verify $\tilde{U}^L(M_{ab}) = (g_{ab}e^{i\lambda_{ab}})^L M_{ab}$.

B7. Minor Point 2: Mirror RB exists for universal gatesets (<https://arxiv.org/abs/2207.07272>).

Response: We thank the referee for pointing out this reference. This work is really a significant extension of Mirror RB to incorporate some non-Clifford gates. But it's still restricted to some particular gate sets, i.e., Clifford gates, arbitrary 1-qubit gates, and 2-qubit controlled Pauli rotation. It cannot be used to benchmark the multi-qubit unitaries presented in the response to A3. Furthermore, Mirror RB is a method to test the holistic performance of a quantum computer using mirrored circuits with randomly sampled cycles composed of 1-qubit and 2-qubit gates. It can estimate the average fidelity of random cycles but not that of a specific structured cycle that our CSB is meant to estimate. Finally, our CSB can measure the information of unitary noise which is however not possible by Mirror RB.

We have modified Table I accordingly to incorporate this reference.

B8. Minor Point 3: The authors' state that "Although its variant, the gate-set tomography, can handle SPAM errors, the experimental costs cannot be reduced." This is certainly true in the presence of arbitrary noise. However, it is not obviously true if you assume some sort of reduced noise model.

Response: We agree with Referee B that the experimental costs could be reduced if we assume noise has some particular structures or properties, such as low rank. We have modified this sentence to mention this point: "the experimental costs cannot be reduced unless assuming noise models with some properties such as low rank [18]".

B9. Minor Point 4: In the supplementary material, the author's refer to the "symmetric group of U ." I am unsure of what this refers to. The provided example looks to be the centralizer of U (although I am often wrong!).

Response: We follow the usage of the symmetric group of a Hamiltonian, i.e., the group whose elements commute with the Hamiltonian. A similar usage occurs also in Ref. [70]. To avoid possible confusion, we modify this phrase as "symmetric group (or centralizer) of U ".

B10. *Errata 1: On page 2, replace "MS" with "Mølmer-Sørensen."*

Response: We have replaced "MS" with "Mølmer-Sørensen".

B11. *Errata 2: On page 2, there is some ambiguity in what the authors' mean when they state "rho is an arbitrary operator." Perhaps "rho is an arbitrary density matrix" is clearer?*

Response: A quantum channel is a special class of linear maps from one operator to another. Its input is of course not restricted to the density matrix. In our CSB, a quantum channel often acts on its eigen-operators which are in general not density matrices. Thus we think this sentence is appropriate.

B12. *Errata 3: On page 3, I am not sure what is meant by "Distracting from the Pauli operator basis."*

Response: Previous benchmarking methods usually measure the noise information under a basis composed of Pauli operators. We think this choice of operator basis is by no means special and suggest using the eigen-operators of the target gate. We have changed this sentence to " Instead of focusing on the Pauli operator basis" to avoid confusion.

LIST OF CHANGES MADE

Based on the questions and suggestions from the Referees, we have made several changes. We use the LaTeX diff tool to generate a file “diff.pdf ” for comparing the difference between the original version and the revised version of the main text. In addition, we also summarize the major changes below.

- (1) We have modified the abstract to highlight that our CSB can be used to benchmark and calibrate global entangling gates.
- (2) In the last sentence of the first paragraph of page 1, we add some words to mention that the experimental cost of GST could be reduced if assuming some special noise models.
- (3) In the introduction, we add a paragraph to emphasize that our CSB can measure the information of unitary noise.
- (4) In Table I, we add a reference for Mirror RB which is extended to some non-Clifford gates, and modify the table entries and caption accordingly.
- (5) We replace “MS gates” with “Mølmer-Sørensen gates”.
- (6) In the introduction, we remove the application of CSB to Grover iteration operator in Grover’s search algorithm. As the ideal eigenvalues of the operator can only be determined if we know the number of solutions to the search problem, which is however usually not known in advance.
- (7) In the text above Eq. (5) on page 3, we add a sentence to explicitly express the assumption on the noisy channel.
- (8) Below Eq. (12) on page 6, we add an in-depth comparison between our CSB and spectral quantum tomography.
- (9) In the section “Ising Hamiltonian evolution” on page 10, we present some important classes of unitary operators frequently used in quantum algorithms or error correction, which are more or less related to the Ising-type of Hamiltonian and satisfy the conditions of scalability of CSB.
- (10) In the bottom left of page 3, we have changed the phrase “Distracting from the Pauli operator basis” to “Instead of focusing on the Pauli operator basis”.
- (11) In the Sec. III of supplementary information, we have changed “symmetric group of U ” to “symmetric group (or centralizer) of U ”.

-
- [1] Maslov, D. & Nam, Y. Use of global interactions in efficient quantum circuit constructions. *New Journal of Physics* **20**, 033018 (2018).
 - [2] van de Wetering, J. Constructing quantum circuits with global gates. *New Journal of Physics* **23**, 043015 (2021).
 - [3] Grzesiak, N., Maksymov, A., Niroula, P. & Nam, Y. Efficient quantum programming using EASE gates on a trapped-ion quantum computer. *Quantum* **6**, 634 (2022).
 - [4] Bravyi, S., Maslov, D. & Nam, Y. Constant-cost implementations of clifford operations and multiply-controlled gates using global interactions. *Phys. Rev. Lett.* **129**, 230501 (2022).
 - [5] Lu, Y. et al. Global entangling gates on arbitrary ion qubits. *Nature* **572**, 363–367 (2019).
 - [6] Figgatt, C. et al. Parallel entangling operations on a universal ion-trap quantum computer. *Nature* **572**, 368–372 (2019).
 - [7] Grzesiak, N. et al. Efficient arbitrary simultaneously entangling gates on a trapped-ion quantum computer. *Nature Communications* **11**, 1–6 (2020).
 - [8] Kassal, I., Jordan, S. P., Love, P. J., Mohseni, M. & Aspuru-Guzik, A. Polynomial-time quantum algorithm for the simulation of chemical dynamics. *Proceedings of the National Academy of Sciences* **105**, 18681–18686 (2008).
 - [9] Jordan, S. P., Lee, K. S. & Preskill, J. Quantum algorithms for quantum field theories. *Science* **336**, 1130–1133 (2012).
 - [10] Li, A. C. Y., Macridin, A., Mrenna, S. & Spentzouris, P. Simulating scalar field theories on quantum computers with limited resources. *Phys. Rev. A* **107**, 032603 (2023).
 - [11] Schlingens, A. W., Head-Marsden, K., Sager-Smith, L. M., Narang, P. & Mazziotti, D. A. Quantum state preparation and nonunitary evolution with diagonal operators. *Phys. Rev. A* **106**, 022414 (2022).
 - [12] Nakata, Y., Hirche, C., Morgan, C. & Winter, A. Unitary 2-designs from random x-and z-diagonal unitaries. *Journal of Mathematical Physics* **58**, 052203 (2017).
 - [13] Nielsen, M. A. & Chuang, I. L. *Quantum computation and quantum information* (Cambridge University Press, 2004), 1 edn.
 - [14] Farhi, E., Goldstone, J. & Gutmann, S. A quantum approximate optimization algorithm. *arXiv preprint arXiv:1411.4028* (2014).
 - [15] Grover, L. K. A fast quantum mechanical algorithm for database search. In *Proceedings of the Twenty-Eighth Annual ACM Symposium on Theory of Computing*, STOC '96, 212–219 (Association for Computing Machinery, New York, NY, USA, 1996).
 - [16] Gilyén, A., Su, Y., Low, G. H. & Wiebe, N. Quantum singular value transformation and beyond: Exponential improvements for quantum matrix arithmetics. *STOC 2019*, 193–204 (Association for Computing Machinery, New York, NY, USA, 2019).
 - [17] Martyn, J. M., Rossi, Z. M., Tan, A. K. & Chuang, I. L. Grand unification of quantum algorithms. *PRX Quantum* **2**, 040203 (2021).
 - [18] Brieger, R., Roth, I. & Kliesch, M. Compressive gate set tomography. *PRX Quantum* **4**, 010325 (2023).

REVIEWERS' COMMENTS

Reviewer #1 (Remarks to the Author):

I would like to thank the authors for their extended comments and answers to my questions, which in particular helped to clarify the differences between interleaved RB and the channel spectrum approach. I am still not completely convinced about the practical relevance of the method. For example the authors mention the case where Trotterization is possible, but then one could equally well just use tomographic or Hamiltonian learning methods to obtain the system parameters locally without additional assumption or the need for diagonalizing the system. However, I do see that the method could be useful in some circumstances.

Reviewer #2 (Remarks to the Author):

No additional comments. Thank you for thoroughly addressing the original feedback. I appreciated the in-depth comparison to spectral tomography as well as the longer discussion of global entangling gates. They both help to highlight the potential impact of the paper.

RESPONSE TO REFEREE A

I would like to thank the authors for their extended comments and answers to my questions, which in particular helped to clarify the differences between interleaved RB and the channel spectrum approach.

Response: We gratefully thank Referee A for reviewing our revised manuscript. We are glad to see that the differences between our CSB and interleaved RB have been clarified.

I am still not completely convinced about the practical relevance of the method. For example the authors mention the case where Trotterization is possible, but then one could equally well just use tomographic or Hamiltonian learning methods to obtain the system parameters locally without additional assumption or the need for diagonalizing the system.

Response: In the case of the Trotterized Hamiltonian evolution unitary operator, we use CSB on each cycle of the Trotterized circuit, which implements the evolution of some terms in the Hamiltonian. These Hamiltonian terms typically have some locality structure or tensor-product structure due to the requirement of efficient circuit implementation, which enables the corresponding ideal unitary operator to be easily diagonalized.

However, tomographic methods require a lot of resources. Although the ideal Trotterized Hamiltonian evolution operator may have tensor product structure, crosstalk noise between qubits may still exist, making the noisy process highly non-local. Doing a full process tomography is of course prohibitive in this case. While performing process tomography locally for each qubit or pair of qubits is feasible (albeit still very resource-expensive), the lack of crosstalk consideration makes it impossible to acquire accurate noise information. Moreover, process tomography usually suffers from SPAM errors.

Hamiltonian learning methods are helpful to identify some unknown parameters in a Hamiltonian or correspondingly some unitary parameters in a quantum gate, which can also be estimated by our CSB method. However, the estimate of unitary parameters can only give some specific information about unitary noise. One can not obtain the average gate fidelity and the stochastic fidelity from the Hamiltonian learning methods. Meanwhile, our CSB can give these three types of noise information in a single experiment.

However, I do see that the method could be useful in some circumstances.

Response: We thank Referee A for the appreciation of the usefulness of our method.

RESPONSE TO REFEREE B

No additional comments. Thank you for thoroughly addressing the original feedback. I appreciated the in-depth comparison to spectral tomography as well as the longer discussion of global entangling gates. They both help to highlight to potential impact of the paper.

Response: We gratefully thank Referee B for reviewing our revised manuscript and the appreciation on the potential impact of our paper.